# KCTD15 acts as an anti-tumor factor in colorectal cancer cells downstream of the demethylase FTO and the m6A reader YTHDF2
Fang-Yuan Zhang[1,5], Lin Wu[2,5], Tie-Ning Zhang ®[3] ✉ & Huan-Huan Chen ®[4] ✉

Potassium Channel Tetramerization Domain Containing 15 (KCTD15) participates in the carcinogenesis of several solid malignancies; however, its role in colorectal cancer (CRC) remains unclear. Here we find that KCTD15 exhibits lower expression in CRC tissues as compared to para-carcinoma tissues. Tetracycline (tet)-induced overexpression and knockdown of KCTD15 confirms KCTD15 as an anti-proliferative and pro-apoptotic factor in CRC both in vitro and in xenografted tumors. N6-methyladenosine (m6A) is known to affect the expression, stabilization, and degradation of RNAs with this modification. We demonstrate that upregulation of fat mass and obesity-associated protein (FTO), a classical m6A eraser, prevents KCTD15 mRNA degradation in CRC cells. Less KCTD15 RNA is recognized by m6A 'reader' YTH N6-Methyladenosine RNA Binding Protein F2 (YTHDF2) in FTO-overexpressed cells. Moreover, KCTD15 overexpression decreases protein expression of histone deacetylase 1 (HDAC1) but increases acetylation of critical tumor suppressor p53 at Lys373 and Lys382. Degradation of p53 is delayed in CRC cells post-KCTD15 overexpression. We further show that the regulatory effects of KCTD15 on p53 are HDAC1-dependent. Collectively, we conclude that KCTD15 functions as an anti-growth factor in CRC cells, and its expression is orchestrated by the FTO-YTHDF2 axis. Enhanced p53 protein stabilization may contribute to KCTD15's actions in CRC cells.

Colorectal cancer (CRC) is the third most common malignancy in the world, with more than 2 million new cases and more than 1 million deaths each year[1,2]. These cases and deaths are related to modifiable risk factors, such as smoking, unhealthy diet, alcohol abuse, decreased physical activity, or being overweight[3]. Colonoscopy and fecal occult blood tests are the current diagnostic methods for CRC[4]. Multiple treatment options for CRC include surgery, radiotherapy, chemotherapy and a combination of these methods improve the survival of CRC patients, but local recurrence, distant metastasis, and final recurrence are still observed in many CRC patients[5,6]. Therefore, there is an urgent need to identify new targets for the effective treatment of CRC.

Potassium (K+) channel tetramerization domain (KCTD) protein family contains 25 members[7]. These proteins commonly function as components of Cullin3 (Cul3)-dependent E3 ligases, and selectively recruit substrates for ubiquitination[8,9]. They are reported to be implicated in a variety of diseases, including neurodegenerative diseases and cancers[7,9]. KCTD15 belongs to the KCTD family and plays an inconsistent role in the carcinogenesis of different cancers. Coppola et al. demonstrated that, in patients with HER2 positive breast cancer, KCTD15 was abnormally overexpressed in the cancerous tissues, and its knockdown inhibited the proliferation of SKBR3 cells, and further sensitized these cells to doxorubicin[10]. A similar tumor-promoting role of KCTD15 was also

¹Department of General Surgery, Shengjing Hospital of China Medical University, Shenyang, China. ²Department of Thoracic Surgery, Shengjing Hospital of China Medical University, Shenyang, China. ³Department of Pediatrics, Shengjing Hospital of China Medical University, Shenyang, China. ⁴Department of Oncology, Shengjing Hospital of China Medical University, Shenyang, China. ⁵These authors contributed equally: Fang-Yuan Zhang, Lin Wu. ✉e-mail: cmuztn@vip.qq.com; chen891107@sina.com

reported in non-solid malignancy B-cell type acute lymphoblastic leukemia by Smaldone and colleagues[11]. However, KCTD15 functions differently in medulloblastoma cells. A previous study from Spiombi et al. showed that the colony formation ability and Edu incorporation of DAOY cells overexpressing exogenous KCTD15 were reduced significantly[12]. The expression data of KCTD15 collected in Gene Expression Profiling Interactive Analysis (GEPIA) database[13] illustrated a lower expression of KCTD15 in CRC tissues than in non-cancer tissues. The controversial role of KCTD15 revealed in different types of cancers and the abnormal downregulation of KCTD15 in CRC specimens provoke us to explore KCTD15's function in CRC.

N6-methyladenosine (m6A) is a commonly RNA modification found in eukaryotes where the N6 position of adenosine is methylated[14]. The m6A enables the modified mRNAs to be recognized by the so-called m6A 'readers', resulting in alterations of RNA translation, splicing, and degradation[14]. High-confidence m6A sites on KCTD15 mRNA are identified via SRAMP online software (http://www.cuilab.cn/sramp)[15], leading us to investigate the association between KCTD15 expression and m6A modification. Fat mass and obesity-associated protein (FTO) is a demethylase that functions as a m6A 'eraser'. In cells derived from CRC patients, low FTO induced a pan-elevation in m6A levels, and promoted *in vivo* tumorigenicity and chemoresistance of CRC cells, suggesting FTO as an anti-CRC factor[16]. FTO deletion-induced m6A modification of certain targeted RNAs makes them easily recognized by m6A 'reader' YTH N6-Methyladenosine RNA Binding Protein F2 (YTHDF2), leading to the degradation of the RNAs[17]. Yang et al.[17] analyzed the gene expression patterns of melanoma cells with or without FTO knockdown via microarray and found that KCTD15 mRNA expression was downregulated upon FTO silencing. Interestingly, data from GSE142825 showed that, in glioblastoma cells, KCTD15 mRNA expression was upregulated after YTHDF2 silencing, suggesting a negative regulatory role of YTHDF2 in KCTD15 mRNA stabilization. The above facts in cancer cells make us propose a hypothesis that KCTD15 may be targeted by the FTO-YTHDF2 axis in CRC cells.

Protein 53 (p53) is a well-known suppressor in varied cancers, including CRC[18]. Histone deacetylase 1 (HDAC1) removes acetyls from p53, thereby accelerating ubiquitination-mediated degradation of p53 protein[19]. Of note, Spiombi et al.[12] reported that KCTD15 was able to decrease the protein expression of HDAC1. We thus focused on the HDAC1-p53 axis to explore KCTD15's function in CRC cells.

The major aims of our study are to 1st explore how KCTD15 affects CRC cell growth and apoptosis and whether the HDAC1-p53 pathway is involved; 2nd whether the abnormal expression of KCTD15 in CRC tissues is associated with m6A modification mediated by FTO and YTHDF2.

## Results

### Bioinformatic analysis of differentially expressed genes (DEGs) in CRC

DEGs ($|\log_2$ (fold change)| >1.0 and $P < 0.05$) overlapped in three datasets (GSE146587, GEPIA-colon adenocarcinoma (COAD) and GEPIA-rectum adenocarcinoma (READ)) were first analyzed. As shown in Fig. 1a, all genes in each dataset were presented in Volcano plots, and the overlapped DEGs were shown in the Venn diagram (Fig. 1b). A total of 1677 DEGs were identified (Fig. 1b). The heatmap showed the expression of these common DEGs in GSE146587 database (Fig. 1c). Gene ontology (GO) analysis illustrated that the common 1677 DEGs were mainly annotated into mitotic nuclear division, nuclear division, organelle fission, and chromosome, centromeric region (Fig. 1d). Kyoto Encyclopedia of Genes and Genomes (KEGG) enrichment showed that the DEGs were mainly enriched in the pathways related to cell growth, such as cell cycle, DNA replication, and p53 pathways (Fig. 1e).

### The expression patterns of KCTD family members

KCTDs are often dysregulated in the progression of multiple cancers[9]. Therefore, we first explored the expression patterns of this family in the above three CRC datasets. The expression of KCTDs was shown in a heatmap (Fig. 2a). Results from GSE146587 showed that KCTD4, 7, 8, 9,

and 15 were significantly decreased in CRC tissues, while KCTD16 was increased (Fig. 2a). GEPIA-COAD displayed that KCTD7, 12, and 15 were downregulated in CRC tissues, while KCTD5 and 14 were highly expressed (Fig. 2a). GEPIA-READ indicated that KCTD1, 7, 12, and 15 were markedly decreased in CRC tissues, whereas KCTD5 and 14 were overexpressed (Fig. 2a). Collectively, KCTD15 (Fig. 2b) and KCTD7 (Supplementary Fig. 1) were commonly decreased in the three datasets.

### KCTD15 inhibits CRC cell proliferation in vitro and in vivo

The decreased KCTD15 expression was also verified in paired normal and CRC tissues (Fig. 2c). The immunohistochemistry (IHC) staining showed that KCTD15 was downregulated in the CRC tissues compared with the para-cancerous tissues (Fig. 2d). Different expression of KCTD15 had no significant impact on age, gender, tumor size, depth of tumor invasion, and differentiation (Table 1). Patients of CRC stages I–II preferred to have low KCTD15 expression. Sample size from the late CRC stages was inadequate (Table 1). Tetracycline (tet)-inducible expression vectors were applied in the study. KCTD15 expression was verified in CRC cells by quantitative realtime PCR (qRT-PCR) and Western Blot (Supplementary Fig. 2a). MTT assay was performed to measure cell viability. We found that KCTD15 overexpression caused a significant decrease in cell viability (Fig. 3a). The decreased 5-Ethynyl-2'-deoxyuridine (EdU)-positive cells (Fig. 3b) and weakened colony formation ability (Fig. 3c) further indicated the impaired cell proliferation post-KCTD15 overexpression. Meanwhile, Ki67 and PCNA protein levels were decreased after KCTD15 upregulation as well (Fig. 3d). KCTD15 silencing promoted the proliferative ability of CRC cells (Fig. 3).

The effects of KCTD15 on cell growth were further verified by in vivo experiments. To induce overexpression or knockdown of KCTD15 in vivo, mice were fed with Tet-containing water when the tumor was visible. The results showed that the induction of KCTD15 expression significantly inhibited tumor growth (Fig. 4a–c). Results of Western Blot demonstrated that the expression of KCTD15 in tumor tissues was successfully induced (Fig. 4e). The effects of KCTD15 on tumor growth were further verified by the detection of Ki67 expression via Western Blot and IHC staining (Fig. 4d–e). Overall, the results demonstrated that KCTD15 suppressed CRC cell proliferation.

### KCTD15 promotes apoptosis in vitro and in vivo

Apoptotic CRC cells were probed by flow cytometry with Annexin V-fluorescein isothiocyanate (FITC) and propidium iodide (PI) double staining. The results indicated that KCTD15 overexpression significantly increased the percentage of apoptotic cells in both HCT116 and LoVo cells (Fig. 5a). The expression of cell apoptosis-related biomarkers, including cleaved caspase 3, cleaved caspase 9, and p53, were increased when KCTD15 was overexpressed (Fig. 5b). The percentage of apoptotic CRC cells under normal conditions was small, and no significant change was observed after knocking KCTD15 down (data not shown). Meanwhile, apoptotic cells within tumor tissues were detected via TUNEL staining, and the results indicated that KCTD15 overexpression induced apoptosis in tumor tissues (Fig. 5c). IHC staining of p53 indicated that KCTD15 overexpression increased p53 expression (Fig. 5d).

### KCTD15 expression is upregulated by FTO overexpression in CRC cells

Six high-confidence m6A sites on KCTD15 mRNA were predicted by SRAMP (http://www.cuilab.cn/sramp), leading us to explore whether m6A modification of KCTD15 affected its expression. We found that the mRNA levels of FTO and KCTD15 were positively correlated in clinical samples from CRC patients (Supplementary Fig. 3; $r = 0.84$; $P < 0.0001$). In addition, the results in Fig. 6a showed that the FTO overexpression significantly increased KCTD15 mRNA level, whilst FTO silencing reduced KCTD15 expression both in HCT116 and LoVo cells. Moreover, FTO overexpression delayed the degradation of KCTD15 mRNA, whereas FTO silencing accelerated this process (Fig. 6b). The six high-confidence m6A sites on

KCTD15 mRNA were shown in Fig. 6c. Primers were designed to amplify the fragments (sequences 1 and 2) near m6A positions (Fig. 6c). RNA immunoprecipitation (RIP) assay showed that FTO bound to KCTD15 mRNA (Fig. 6d). To clarify whether FTO can regulate m6A modification of KCTD15, m6A RNA immunoprecipitation (MeRIP)-qPCR was then conducted. As shown in Fig. 6e, a basal m6A modification was detectable on KCTD15 mRNA. FTO overexpression attenuated KCTD15 m6A modification (Fig. 6e). To further explore how m6A modification affected KCTD15 mRNA expression, the base A within the m6A motif was replaced by a base C. Both wild-type and mutant fragments were inserted into the

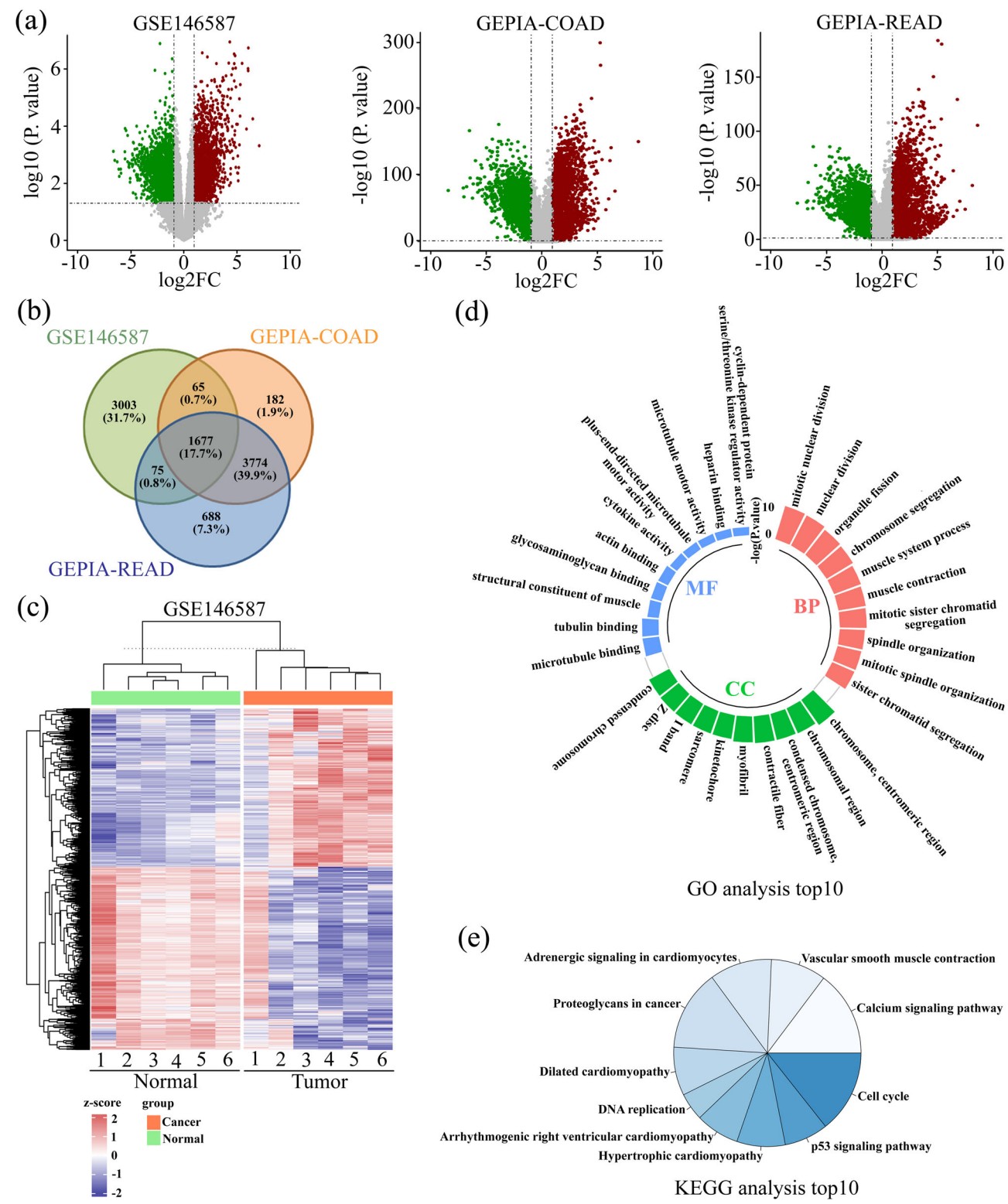

**Fig. 1 | Bioinformatics analysis of differentially expressed genes (DEGs) in CRC. a** DEGs from the three profiles were exhibited as Volcano plots. Genes with |log₂ (fold change)| >1.0 and *P* < 0.05 were considered DEGs. **b** Venn diagram was used to present the common DEGs among the three datasets. **c** The heatmap showed the expression of these common DEGs in the GSE146587 database. **d, e** GO enrichment and KEGG analysis were performed, and the TOP10 items were listed.

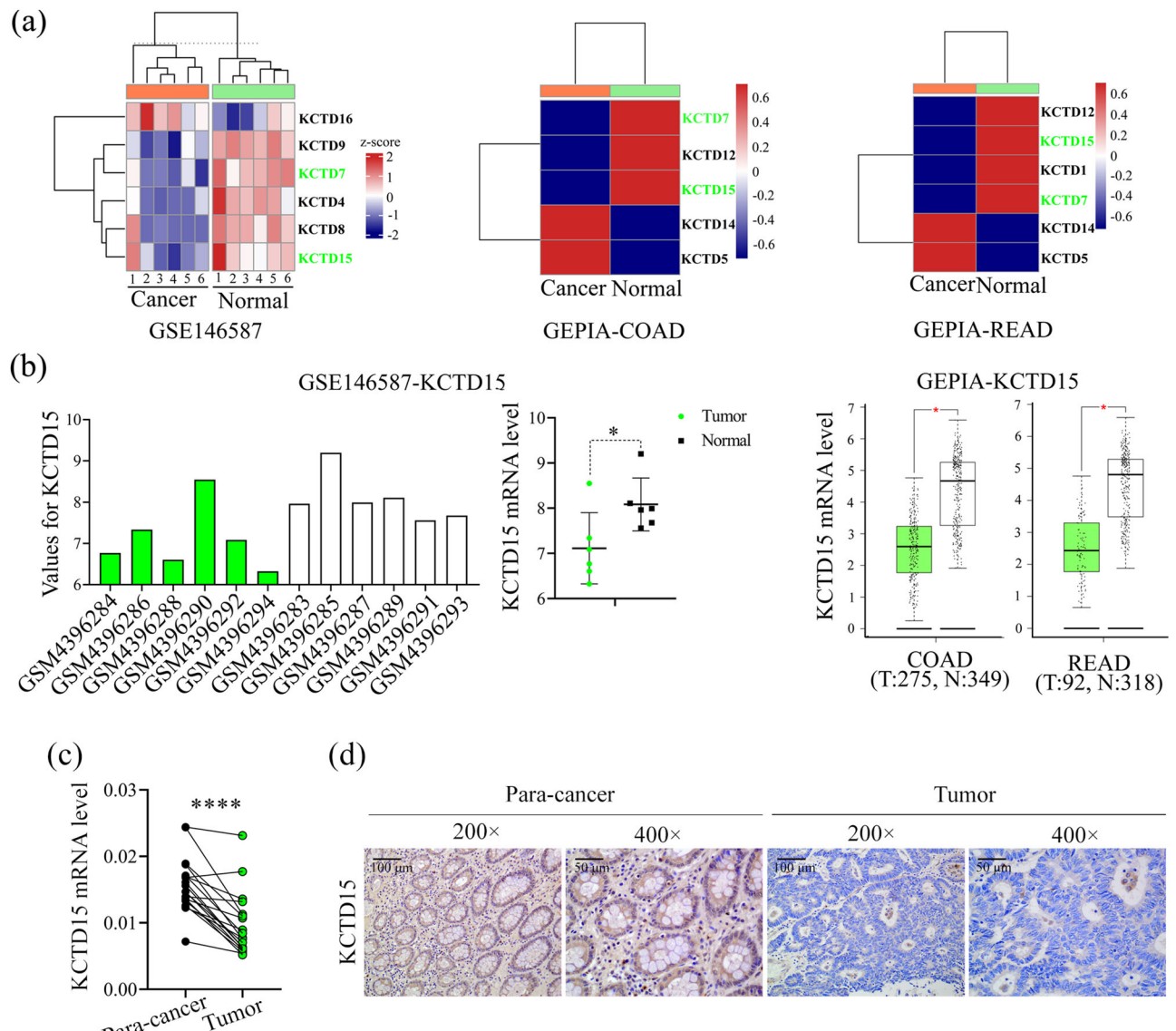

**Fig. 2 | The expression of KCTD family members. a** The expression of KCTD family members was shown by a heatmap. **b** Expression values of KCTD15 in each dataset were shown. **c** KCTD15 mRNA expression was decreased in CRC tissues as compared with paired normal tissues. **d** Representative IHC staining images of KCTD15 expression (200 × and 400 ×). *$P < 0.05$ and ****$P < 0.0001$.

---

pmir-GLO dual luciferase vectors and transfected into HCT116 cells. The results showed that the mRNA of wild-type KCTD15 was more stable when FTO was overexpressed (Fig. 6f).

### The m6A 'reader' YTHDF2 mediates the effect of FTO on the mRNA stability of KCTD15

RNAs with modified m6A modification will be recognized by multiple m6A 'readers'[20]. YTHDF1, 2, and 3 are valid and well-studied m6A 'readers' with conservative m6A binding domains. We determined the effects of those three 'readers' on KCTD15 mRNA expression by knocking each of them down in CRC cells. Our findings indicated that the silencing of YTHDF2, but not YTHDF1 or 3, increased KCTD15 mRNA expression (Fig. 7a).

We assumed that FTO affected the stability of KCTD15 RNA in a YTHDF2-dependent manner (Fig. 7b). To verify this hypothesis, RIP-qPCR was performed and the results showed that YTHDF2 bound to KCTD15 mRNA (Fig. 7c). Moreover, less YTHDF2 was found to interact with the mRNA of KCTD15 in cells overexpressing FTO (Fig. 7c). More YTHDF2 was identified to bind to KCTD15 mRNA in cells transfected with FTO siRNA (Fig. 7c). By analyzing the mRNA expression of KCTD15 in

HCT116 cells, we found that FTO silencing was unable to significantly upregulate KCTD15 expression when the m6A reader YTHDF2 was knocked down (Fig. 7d). No significant alteration in FTO expression was observed after YTHDF2 silencing (Fig. 7e). Based on the above, we demonstrated that KCTD15 mRNA expression was associated with FTO/YTHDF2-mediated m6A modification.

### KCTD15 inhibits CRC progression by increasing the protein stability of p53 in a HDAC1-dependent manner

It has been reported that KCTD15 is an upstream negative regulator of HDAC1[12], and that the removal of acetyls from p53 by HDAC1 can accelerate p53 degradation through the ubiquitin-proteasome system[19]. We thus firstly examined the effect of KCTD15 on the acetylation of p53 at Lys373 and Lys382 sites, two sites targeted by ubiquitin ligases[19]. The results showed that KCTD15 overexpression decreased HDAC1 protein level, increased total p53 expression, and enhanced its acetylation at Lys373 and Lys382 sites in two CRC cells (Fig. 8a). Co-immunoprecipitation (Co-IP) results showed no protein interaction of KCTD15 and HDAC1 in CRC cells (Fig. 8b). To further determine whether HDAC1 was involved in KCTD15's

**Table 1 | Correlation between KCTD15 expression and clinicopathological parameters in colorectal cancer patients**

| Parameters | n | KCTD15 expression | | p-value |
|---|---|---|---|---|
| | | High (n = 25) | Low (n = 100) | |
| Age (years) | | | | 0.53 |
| ≥60 | 62 | 11 | 51 | |
| <60 | 63 | 14 | 49 | |
| Gender | | | | 0.92 |
| Male | 86 | 17 | 69 | |
| Female | 39 | 8 | 31 | |
| Tumor size (cm) | | | | 0.95 |
| ≤3 | 23 | 5 | 18 | |
| >3 | 102 | 20 | 82 | |
| TNM stage | | | | 0.04 |
| I–II | 115 | 20 | 95 | |
| III | 10 | 5 | 5 | |
| Depth of tumor invasion | | | | 0.79 |
| T1–T2 | 62 | 13 | 49 | |
| ≥T3 | 63 | 12 | 51 | |
| Differentiation | | | | 0.23 |
| Poor | 14 | 5 | 9 | |
| Well/Moderate | 111 | 20 | 91 | |

*P < 0.05.

regulatory effects on p53, HDAC1 was forced to overexpress in HCT116 cells together with KCTD15. The results in (Fig. 8c–f). showed that the re-expression of HDAC1 disrupted p53 expression, inhibited cell apoptosis, and enhanced cell survival.

Moreover, KCTD15 overexpression reduced the ubiquitination of p53 (Fig. 9a) and the half-life of p53 protein was increased from 26.23 min to 80.95 min after the overexpression of KCTD15 (Fig. 9b). Additionally, the reduced cell survival and enhanced cell apoptosis induced by KCTD15 overexpression were partly reversed by p53 silencing (Fig. 9c–g). The results suggested that KCTD15 inhibited CRC progression by increasing the protein stability of p53 in a HDAC1-dependent manner.

## Discussion

Cytotoxic chemotherapy and immunotherapy, and their combination are currently available treatments for CRC[21]. However, the therapeutic effects are still not optimistic. In recent years, many researchers have devoted themselves to exploring novel biomarkers of carcinogenesis. Biomarkers not only identify the susceptibility or early stage of cancers but may also accelerate the diagnosis and guide the therapeutic strategies. It is even possible to develop personalized treatment plans based on specific biomarkers[22]. Addition of biological agents targeting valid molecules involved in CRC progress, such as vascular endothelial growth factor (VEGF) and epidermal growth factor (EGF), to CRC traditional chemotherapies further improves the survival of CRC patients[23]. Therefore, looking for novel candidates that participated in CRC will provide new ideas for the treatment of CRC. In this study, we demonstrated that KCTD15 expression was significantly decreased in the CRC tissues, and that its mRNA stability was regulated by FTO/YTHDF2-mediated m6A modification. Moreover, KCTD15 inhibited CRC cell growth and triggered apoptosis via upregulating p53 expression.

Common 1677 DEGs identified from three CRC datasets were mainly enriched in cell cycle, DNA replication, proteoglycans in cancer, p53 signaling pathway and calcium signaling pathway. Among them, cell cycle, p53 signaling pathway, and DNA replication are closely related to the growth of cancer cells. The adhesion and migration of cancer cells are regulated by proteoglycans, such as heparan sulfate proteoglycans[24]. Abnormal calcium signaling pathway contributes to the malignant behaviors of cancer cells[25,26]. Such annotation results suggest that the identified DEGs may participate in CRC development at multiple levels. We also found that these DEGs were involved in muscle contraction and cardiomyopathy. Although such findings may not be related to our study, we still believe that others may find it interesting.

Each member of KCTD family has a conserved N-terminal domain and a variable C-terminal domain[27]. The N-terminal domain contains a conserved BTB motif, which is a multifunctional protein-protein interaction motif that promotes homologous dimerization or heterodimerization. At present, various functions of KCTDs have been reported, including transcriptional inhibition[28], cytoskeletal regulation[29], and interaction with ubiquitin ligase complex[30]. Members of this family are differentially expressed in cancerous tissues, and they modulate several specific oncogenic pathways such as PI3K/AKT, NF-κB and Wnt/β-catenin signaling pathways[9]. Therefore, this study focused on KCTDs in CRC progression. Our analysis results showed that both KCTD7 and 15 expression levels were downregulated in CRC tissues as compared to the non-cancer tissues. The mutation or defect of KCTD7 protein is reported to be associated with neurodegenerative disorder myoclonus epilepsy[31,32]. Little is known about its role in cancer. Only two previous literatures derived from clinical data mining showed that the alternative splicing of KCTD7 had a prognostic value in glioblastoma and lung adenocarcinoma[33,34]. However, the precise mechanisms remain unknown. Although the present study focused on KCTD15, we also plan to investigate whether KCTD7 affects CRC progress in the future.

We here demonstrated that KCTD15 inhibited cell proliferation and induced apoptosis in CRC cells and xenografts. The anti-tumor role of KCTD15 was also observed in medulloblastoma[12], but not in breast cancer and B-cell acute lymphoid leukemia[10,11]. These studies suggested the function of KCTD15 may vary among different cancers. We here aimed to elucidate how KCTD15 affected the growth of CRC cells and will explore its role in CRC metastasis.

We analyzed the expression of KCTD15 in 125 CRC samples via IHC assay and classified the corresponding patients into KCTD15-high and KCTD15-low groups. Under the hypothesis that KCTD15 restricted the growth of CRC cells, we assume that CRC patients at early TNM stages have high KCTD15 levels. Unexpectedly, we found patients from stages I–II had preferentially low KCTD15 levels. The expression pattern of KCTD15 in CRC patients with late stages was unclear since only 10 patients of TNM stage III were recruited. Such an inexplicable phenomenon may be that we failed to collect clinical samples from all TNM stages, and the number of early and late stages varied a lot. We extracted data regarding the survival of CRC patients from the online database UALCAN (http://ualcan.path.uab.edu). The results showed that patients with higher KCTD15 expression had a better prognosis (data not shown), though not reaching a statistical difference. KCTD15 and its family members are the current research emphases of our group. We will keep collecting clinical samples in our hospital, conduct a follow-up study of these CRC patients, and share the obtained data in the future.

m6A modification on RNA is one of the reversible forms of epigenetic modification in mammalian cells. m6A modification was controlled by the 'writers', 'erasers', and 'readers' to add, delete, and recognize m6A, respectively. 'Writers' are m6A methyltransferases, including methyltransferase-like 3 (METTL3), methyltransferase-like 14 (METTL14), and Wilms' tumor 1-associating protein (WTAP). 'Erasers' stand for m6A demethylases, such as FTO and AlkB homolog 5 (ALKBH5). The 'readers', such as YTHDF1, 2, 3, can identify the m6A site and further regulate RNA splicing, nuclear output, translation regulation, and decay[35]. FTO was downregulated in CRC tissues and inhibited the malignant behaviors of CRC cells through its m6A demethylase activity[36]. Our results revealed a positive correlation between FTO and KCTD15 expression in CRC tissues. In addition to upregulating the mRNA expression of KCTD15 in HCT116 and LoVo cells, FTO overexpression also attenuated m6A modification of KCTD15. Such findings suggest that FTO-mediated m6A demethylation of KCTD15 may prevent its mRNA degradation. YTHDF3 commonly collaborates with YTHDF1 to

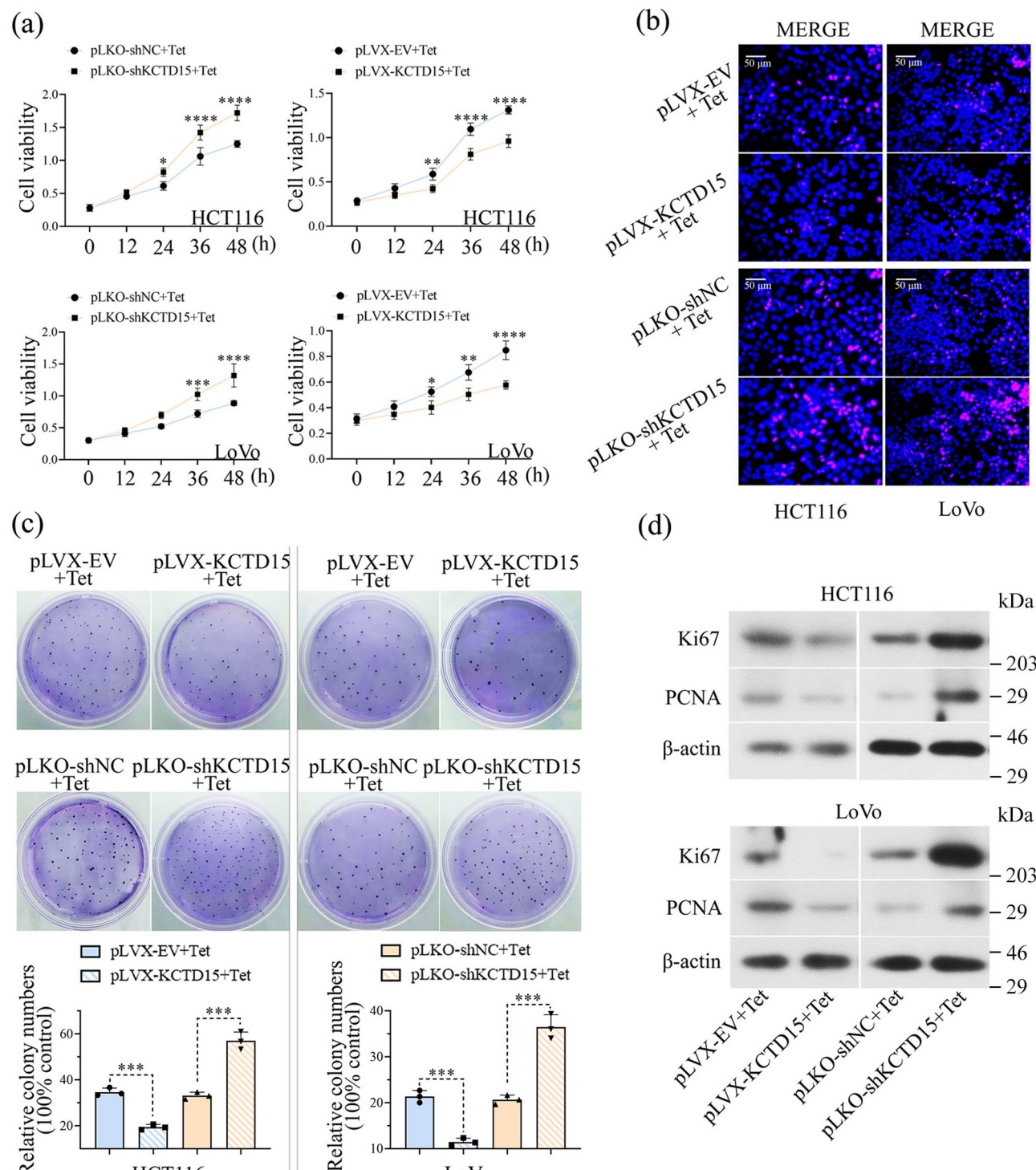

**Fig. 3 | KCTD15 inhibits cell proliferation of CRC cells. a** MTT assay was performed to detect cell viability. **b** DNA synthesis was measured by EdU staining (400 ×). **c** Representative images of plate cloning formation assay. **d** The protein levels of Ki67 and PCNA (cell proliferative markers) were measured by Western Blot. *$P < 0.05$, **$P < 0.01$, ***$P < 0.001$, and ****$P < 0.0001$. $n = 3$ per group.

promote the translation of methylated RNA, while YTHDF2 accelerates the RNA decay[37]. In melanoma cells, FTO increased the RNA stability of programmed cell death 1 ligand 1 (PD-L1) by inducing m6A demethylation, and prevented PD-L1 from being recognized by YTHDF2[38]. In pancreatic cancer cells, although FTO still inhibited the m6A modification of its target tissue factor pathway inhibitor-2 (TFPI2), unlike PD-L1, the mRNA expression of TFPI2 was decreased because less TFPI2 was bound by YTHDF1[39]. These earlier studies suggest that the effects of FTO-mediated

m6A de-modification on RNA expression depend on the m6A readers. In this study, YTHDF2, but not YTHDF1 or 3, reduced the expression of KCTD15. Our study confirmed that the mRNA stability of KCTD15 is orchestrated by FTO-YTHDF2. Whether other 'writers' or 'erasers' participate in the regulation of KCTD15 needs to be further explored.

TP53 mutations occur in approximately 40–50% of CRC[40], and mutant TP53 may encode inactive p53[41]. HCT116 and LoVo cells with no TP53 mutation were used in this study to ensure that the endogenous p53

function as an apoptotic inducer. A previous study from Spiombi et al. demonstrated that KCTD15 reduced HDAC1 protein expression without interacting with HDAC1 in medulloblastoma cells[12]. They found that

KCTD15 was able to directly interact with KCASH2 (KCTD Containing-Cullin Adaptor 2)/KCTD21 through its BTB domain, and this interaction led to an increase in the stability of KCTD21 and accelerated the degradation

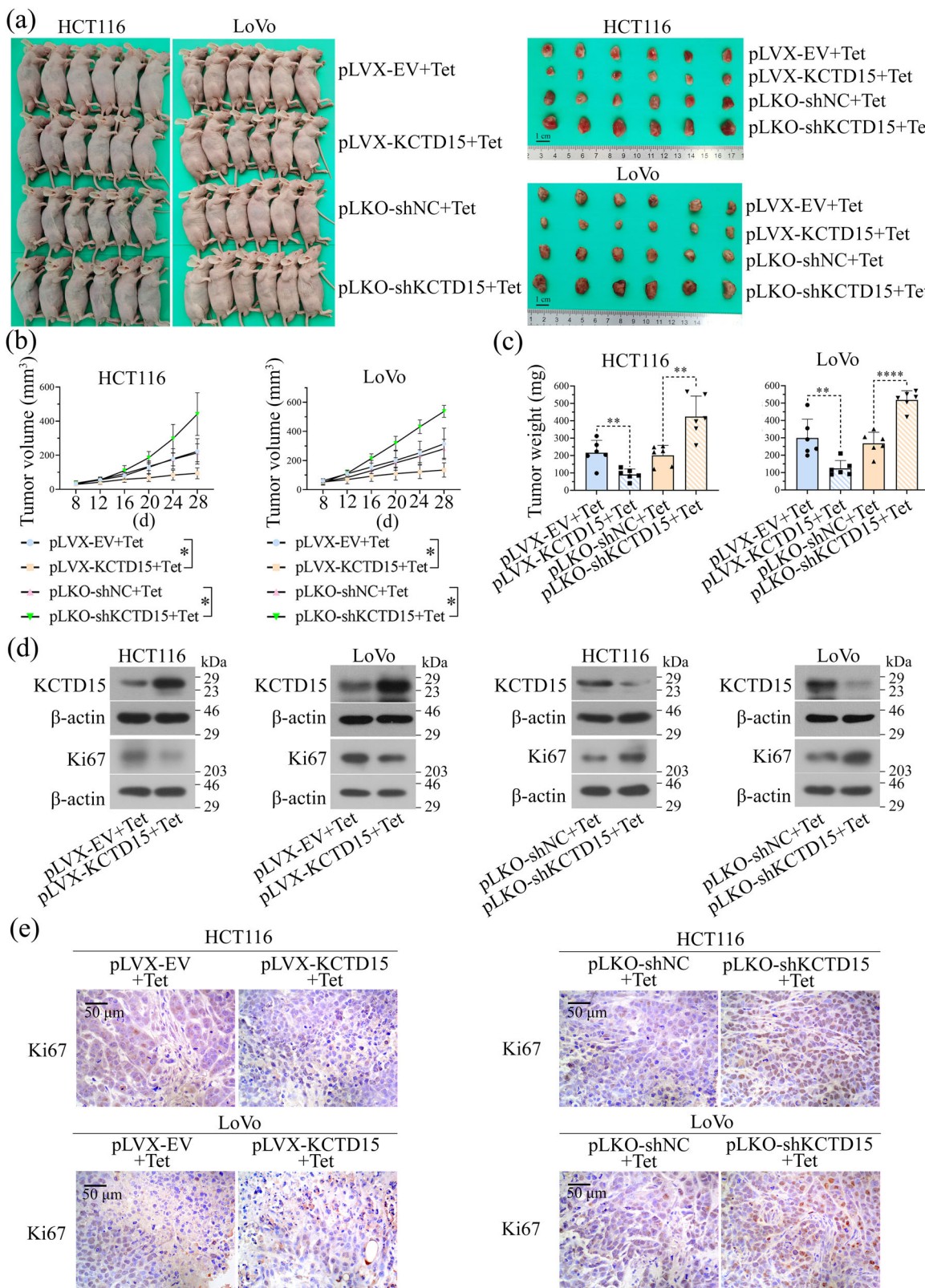

**Fig. 4 | KCTD15 inhibits CRC tumor growth in vivo. a** Images of nude mice and tumor tissues were provided. **b** Tumor volumes were measured at indicated time points. **c** Tumor weights of nude mice. **d** The expression of KCTD15 and Ki67 in tumor tissues was measured by Western Blot. **e** Ki67 expression in tumor tissues was measured by IHC staining (400 ×). *$P < 0.05$, **$P < 0.01$, and ****$P < 0.0001$. $n = 6$ per group.

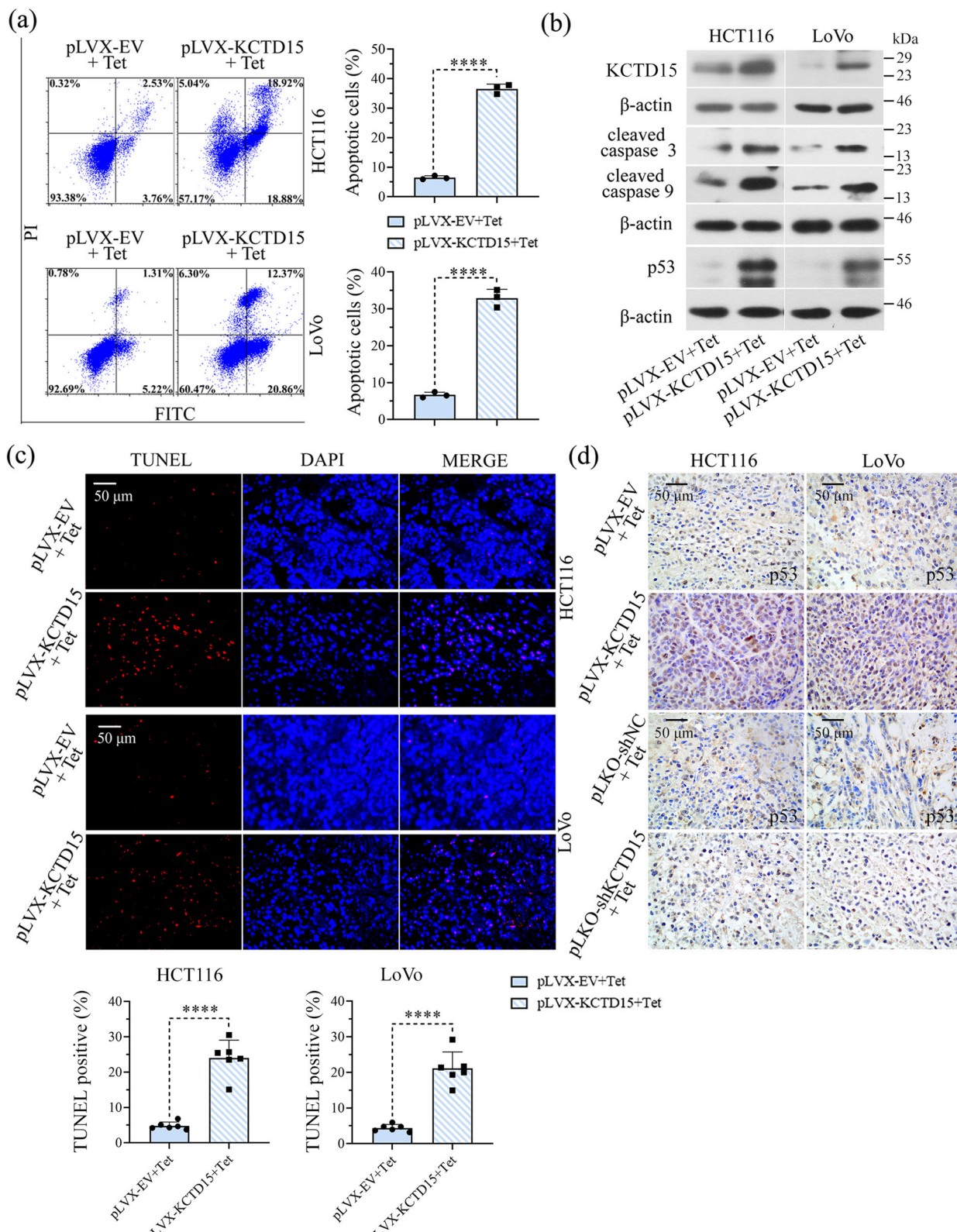

**Fig. 5 | KCTD15 induces cell apoptosis. a** Apoptotic cells were stained with Annexin V-fluorescein isothiocyanate (FITC)/propidium iodide (PI) and analyzed on a flow cytometry. **b** Protein expression of KCTD15 and apoptosis-related markers, including cleaved caspase 3, cleaved caspase 9, and p53, were detected by Western Blot. **c** Apoptosis in tumor tissues was measured by TUNEL staining (400 ×). **d** IHC images of p53 positive cells within the tumor masses (400 ×). ****$P < 0.001$. **a, b**: $n = 3$ per group; **c, d**: $n = 6$ per group.

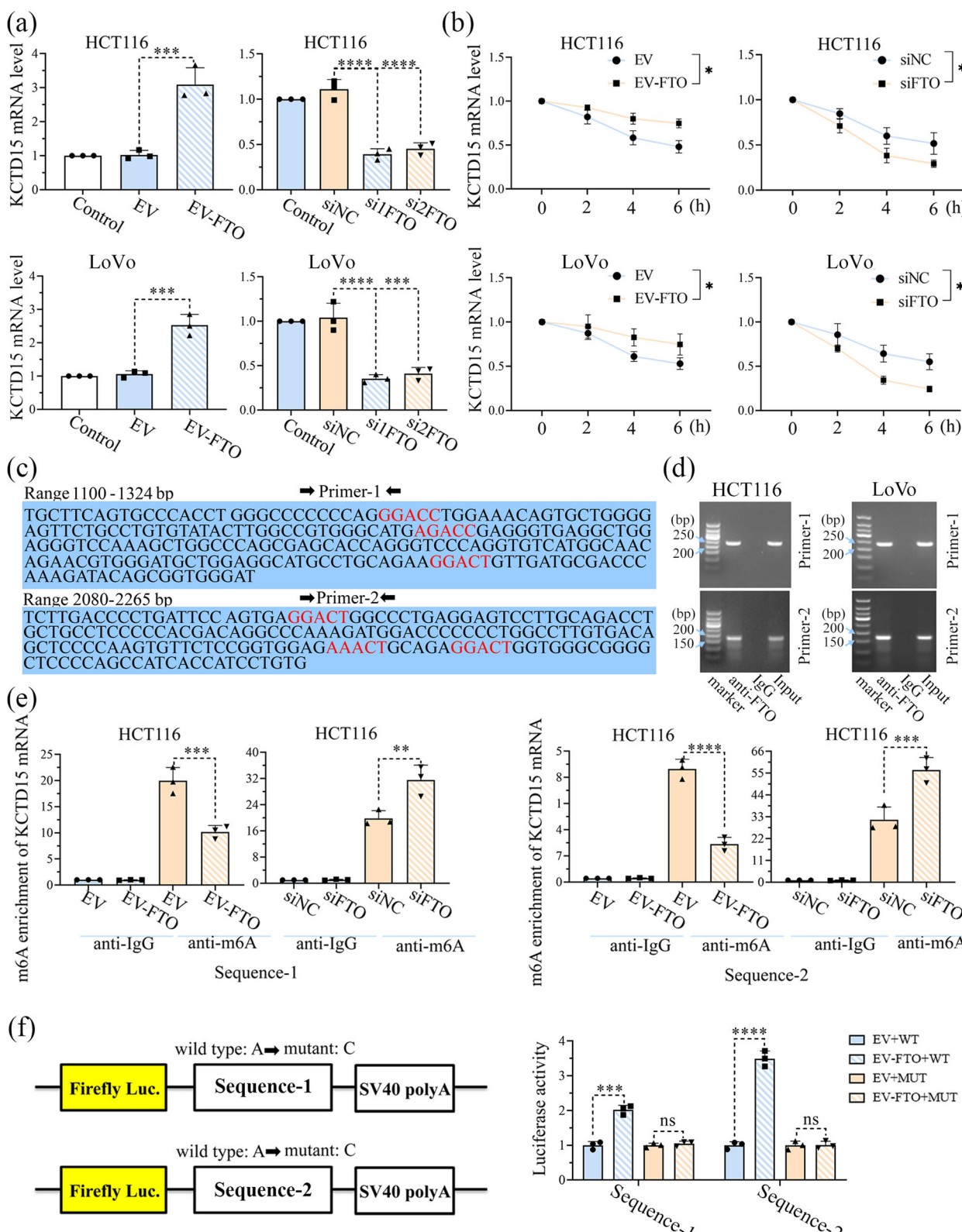

**Fig. 6 | KCTD15 expression is upregulated by FTO through m6A demethylation in CRC cells. a** The mRNA expression of KCTD15 mRNA was determined by qRT-PCR in HCT116 and LoVo cells. **b** KCTD15 mRNA stability at different time points was measured by qRT-PCR. **c** Primers were designed to amplify the fragments (sequences 1 and 2) near m6A positions of KCTD15. **d** Binding of FTO to KCTD15

RNA was measured by RIP-PCR. **e** MeRIP-qPCR was conducted to analyze the m6A demethylation of KCTD15 RNA. **f** To further explore the how m6A modification affected KCTD15 mRNA expression, base A within the m6A motif was replaced by base C. *$P < 0.05$, **$P < 0.01$, ***$P < 0.001$, and ****$P < 0.0001$. $n = 3$ per group.

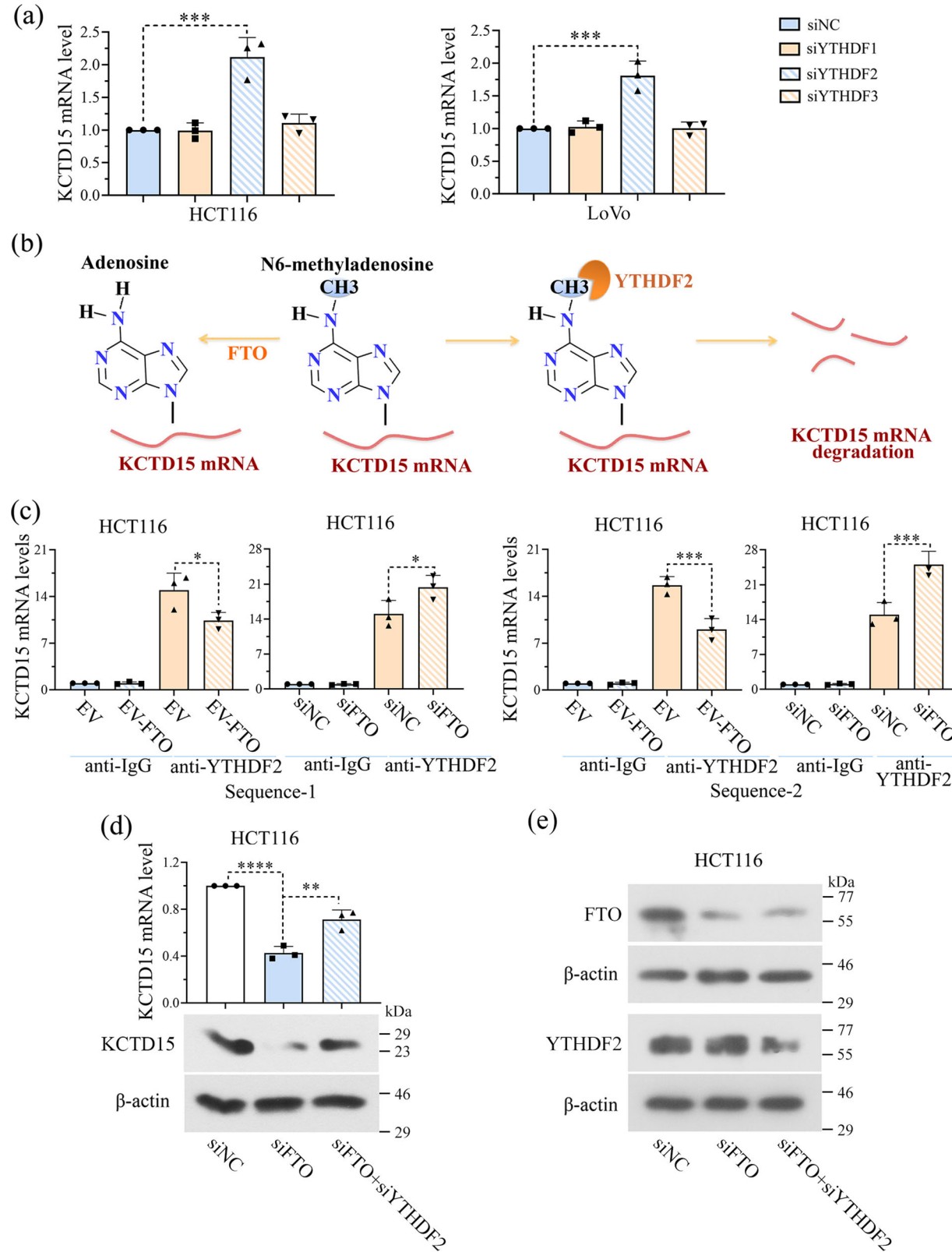

**Fig. 7 | The m6A 'reader' YTHDF2 mediates the effect of FTO on the mRNA stability of KCTD15. a** The effects of YTHDF1, 2, and 3 on KCTD15 mRNA levels were measured by qRT-PCR in HCT116 and LoVo cells. **b** Hypothetical mechanisms regarding to FTO/YTHDF2-mediedated m6A modification of KCTD15. **c** Binding of YTHDF2 to KCTD15 RNA was measured by RIP-qPCR. **d** KCTD15 expression was measured by qRT-PCR and Western Blot. **e** The protein expression levels of FTO and YTHDF2 were detected by Western Blot. $*P < 0.05$, $**P < 0.01$, $***P < 0.001$, and $****P < 0.0001$. $n = 3$ per group.

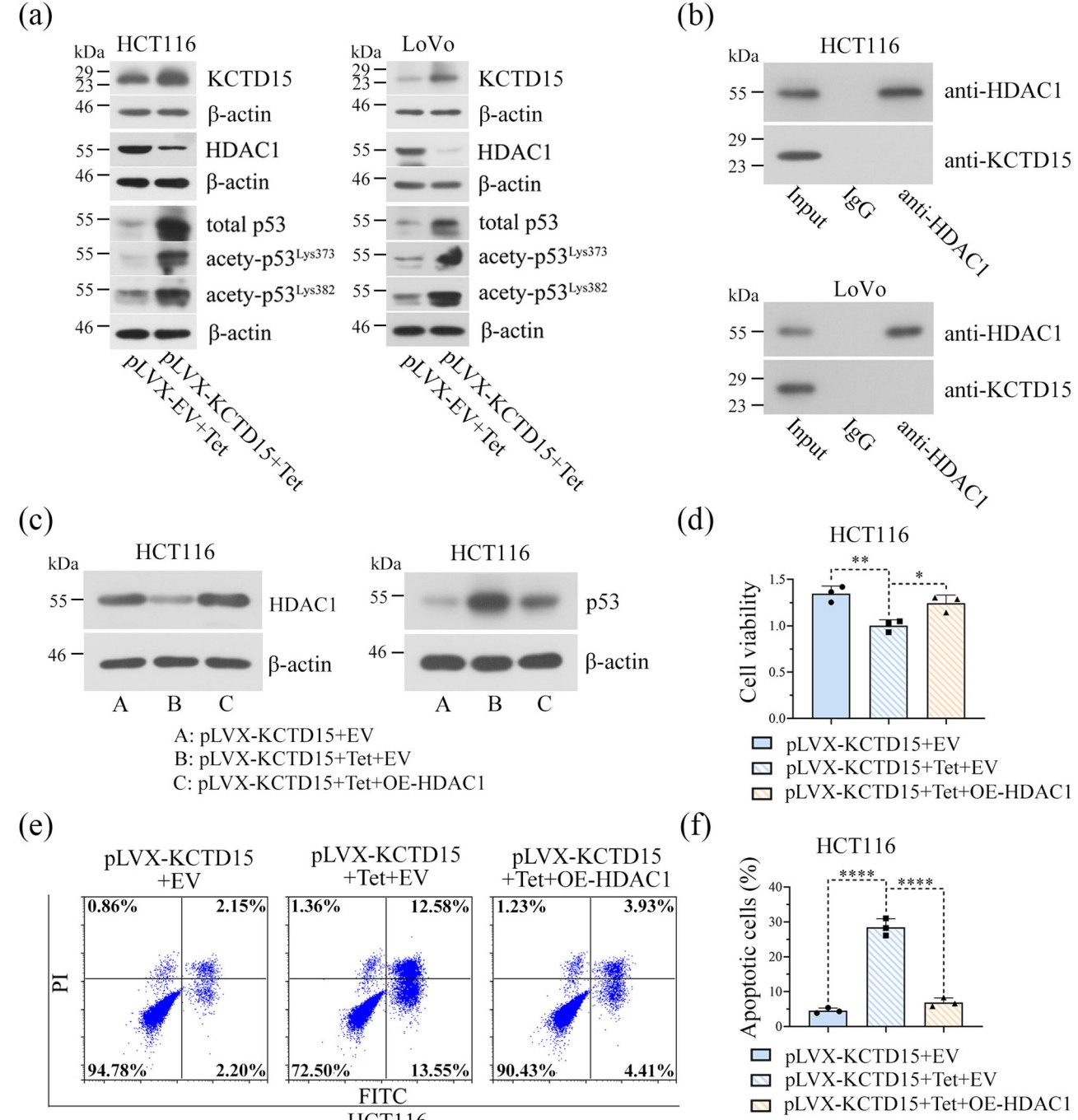

**Fig. 8 | HDAC1 overexpression partly reverses cellular behavior alterations induced by KCTD15 overexpression. a** The protein levels of KCTD15, HDAC1, total and acetylated p53 were determined via Western Blot. **b** The interaction of HDAC1 and KCTD15 was detected by Co-IP. **c** The protein expression levels of HDAC1 and p53 were detected by Western Blot. **d** MTT assay was performed to detect cell viability. **e, f** Apoptotic cells were stained with Annexin V-fluorescein isothiocyanate (FITC)/propidium iodide (PI), and analyzed on a flow cytometry. *$P < 0.05$, **$P < 0.01$ and ****$P < 0.0001$. $n = 3$ per group.

of HDAC1[12]. Given the fact that HDAC1 is a pivotal regulator of p53 deacetylation[19], we here analyzed the protein levels of HDAC1, total p53 and acetylated-p53 in CRC cells. Our findings revealing a downregulation of HDAC1 protein expression in HCT116 and LoVo cells overexpressing KCTD15 were consistent with the findings reported by Spiombi et al[12]. Lysine residues of p53 that can be acetylated often also overlap with those that can be ubiquitylated. Ito et al. demonstrated that HDAC1 promoted p53 deacetylation, and further accelerated its ubiquitylation and degradation[19]. Post-translational modification of proteins may not function independently, and crosstalk between different modifications is one major

way to affect protein expression. Herein, we also found that KCTD15 overexpression enhanced p53 acetylation and upregulated its expression in a HDAC1-dependent manner. Moreover, neither HDAC1 overexpression nor TP53 inhibition completely abolished the apoptosis of CRC cells induced by KCTD15, suggesting an involvement of other mechanisms underlying KCTD15's pro-apoptotic function. We plan to perform high-throughput proteomics to comprehensively explore if there are other molecules involved in KCTD15's actions in CRC cells.

Collectively, we demonstrate that KCTD15 expression is significantly downregulated in CRC tissues, and that its RNA expression and

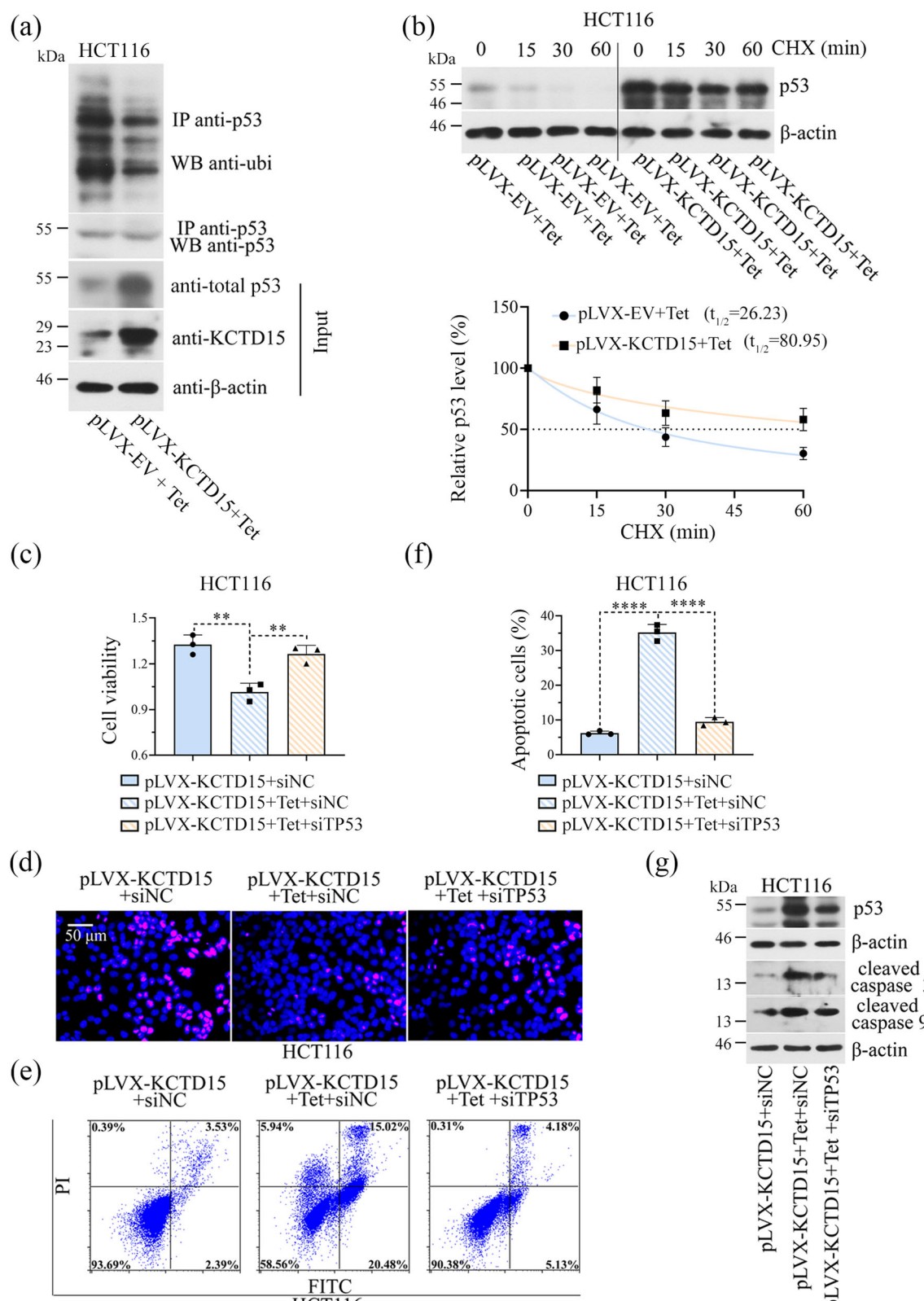

**Fig. 9 | KCTD15 increases the protein stability of p53 by inhibiting HDAC1.**
**a** Ubiquitinated p53 was probed via Co-IP in HCT116 cells. **b** HCT116 cells were treated with 50 μg/mL cycloheximide (CHX), and at the indicated time points, cells were harvested to analyze p53 protein levels via Western Blot. The degradation curve of p53 was shown below. **c** MTT assay was performed to detect cell viability. **d** DNA synthesis of HCT116 cells was measured by EdU staining (400 ×). **e, f** Apoptotic cells were stained with Annexin V-fluorescein isothiocyanate (FITC)/propidium iodide (PI) and analyzed on a flow cytometry. **g** The protein expression levels of p53, cleave caspase 3 (c-caspase 3), and cleaved caspase 9 (c-caspase 9) were detected by Western Blot. **\*\****P* < 0.01 and **\*\*\*\****P* < 0.0001. *n* = 3 per group.

stabilization are regulated by FTO/YTHDF2-mediated m6A modification. KCTD15 exerts it pro-apoptotic effects at least by inducing p53 expression in a HDAC1-dependent way.

## Methods

### Identification of DEGs in CRC tissues and enrichment analysis
A CRC gene expression profile (GSE146587) was downloaded from Gene Expression Omnibus (GEO). Two datasets regarding COAD and READ from The Cancer Genome Atlas (TCGA) were analyzed via GEPIA. There were 6 CRC and 6 normal tissues in GSE146587, 275 tumor and 349 normal tissues in TCGA-COAD datasets, and 92 tumor and 318 normal tissues in TCGA-READ. Genes with $|\log_2$ (fold change)$| > 1.0$ and $P < 0.05$ were considered DEGs. DEGs in each dataset were presented in volcano plots, and the overlapped DEGs were shown in the Venn diagram. GO and KEGG were used to analyze the potential functions of the identified DEGs.

### Clinical samples
All ethical regulations relevant to human research participants were followed. One hundred and twenty-five CRC slices (86 males and 39 females, median age was 59.5, ranged 26–81), and 19 paired fresh normal and CRC specimens (9 males and 10 females, median age was 67, ranged 56–86) were collected from Shengjing Hospital of China Medical University. The informed consents were obtained from all participants. All experiments were approved by the Ethics Committee of Shengjing Hospital of China Medical University (No. 2022PS965K).

### Cell lines and culture
HCT116 and LoVo CRC cell lines were purchased from iCell (Shanghai, China). HCT116 cells were grown in McCoy's 5 A medium (Servicebio, Wuhan, China), while LoVo cells were cultured in F12K medium (iCell) with 10% fetal bovine serum (FBS, Tianhang Biotechnology, Zhejiang, China). Cells were incubated in a 37 °C humidified incubator with 5% $CO_2$.

### Gene knockdown and overexpression
To achieve *KCTD15* knockdown, shRNA specifically targeting *KCTD15* mRNA was inserted into Tet-pLKO-puro lentivirus vector (Fenghui Biotech, Changsha, China) to generate the recombinant Tet-pLKO-puro-shKCTD15 vector. A nonspecific scramble sequence was used as a control, named Tet-pLKO-puro-shNC. Inducible overexpression of *KCTD15* was achieved by using the pLVX-TetOne-Puro lentivirus vector (Fenghui Biotech). The encoding sequence of *KCTD15* was cloned into pLVX-TetOne-Puro to generate a recombinant pLVX-TetOne-Puro-KCTD15 vector. The empty vector served as a control, named pLVX-TetOne-Puro-EV. Cells infected with the above-mentioned lentiviruses were then selected by 1.5 µg/mL puromycin (Macklin, Shanghai, China) to generate stable cell clones. To induce *KCTD15* silencing or overexpression, cells were treated with tetracycline (Tet, 2 µg/mL) at the time point indicated.

Cells were first inoculated into a 6-well plate ($4 \times 10^5$ cells each well) and cultured for 24 h, and then transfected with siRNAs specifically targeting *FTO*, *YTHDF1*, *YTHDF2*, *YTHDF3*, or *TP53* by using lipofectamine 3000 (Invitrogen, Carlsbad, CA, USA) according to the manufacturer's instructions. A nonspecific scramble sequence (siNC) was used as a control. For *FTO* overexpression, the coding sequence of *FTO* was inserted into the pcDNA3.1 vector (Youbio Company, Hunan, China). siRNAs were purchased from General Biology Co., Ltd. (Anhui, China). The sequence information of shRNA and siRNA was provided in Supplementary Table. S1.

### qRT-PCR
Total RNAs were isolated from cells or tissues by using TRIpure lysate (BioTeke, Beijing, China). RNA concentration was measured by an ultraviolet spectrophotometer (NANO 2000, Thermo Scientific, Waltham, MA, USA). cDNA was synthesized using BeyoRT II M-MLV reverse transcriptase (Beyotime, Shanghai, China, 1 µL), RNase inhibitor (Sangon Biotech, Shanghai, China, 0.5 µL), 5 × PCR Buffer (4 µL), and dNTP mixture (2.5 mM each, 2 µL). The following protocol was used for reverse transcription: 25 °C

for 10 min, 42 °C for 50 min, and 80 °C for 10 min. Then, the mRNA expression of targeted genes was quantified via qRT-PCR detection system. A 20-µL reaction system contained 10 µL 2×Taq PCR MasterMix (Solarbio, Beijing, China), 0.3 µL SYBR Green (Solarbio), 1 µL cDNA template, specific primers for the target genes (total 1 µL). The following thermocycling conditions were as follows: 95 °C for 4 min, 40 cycles of 95 °C for 10 s, 60 °C for 10 s, and 72 °C for 15 s. Data were analyzed by the $2^{-\Delta\Delta Ct}$ method and β-actin was used as the internal control to normalize the mRNA expression. Primers used in qRT-PCR were synthesized by General Biology Co., Ltd (Anhui, China), and the primer sequences were provided in Supplementary Table. S2.

To determine the mRNA stabilization of KCTD15, CRC cells were treated with the transcription inhibitor Actinomycin D (ActD, 5 µg/mL) for 0, 2, 4, and 6 h.

### Western Blot
Proteins were extracted using Western and IP cell lysates (Beyotime) supplemented with 1 mM phenylmethanesulfonyl fluoride (Beyotime). A BCA Protein Assay Kit (Beyotime) was used to quantify the protein concentration according to the manufacturer's protocols. Equal amounts of protein (20–40 µg) were separated by SDS-PAGE and transferred on polyvinylidene difluoride (PVDF) membranes (Millipore, Billerica, MA, USA). The PVDF membranes were blocked with 5% non-fat milk in TBST, and then were incubated with primary antibodies overnight at 4 °C followed by the incubation of corresponding secondary antibody for 45 min at 37 °C. The protein bands were visualized with chemiluminescence reagent and the optical density value of the target bands was analyzed by Gel-Pro-Analyzer software. The antibody details were provided in Supplementary Table. S3. During the experiments, the membranes were cropped according to the molecular weights of proteins before incubating primary antibodies, and the neighboring size markers of blots in the Supplementary Figs. were marked according to the position of the cropped membranes.

### MTT assay
Cells were seeded into 96-well plates ($5 \times 10^3$ cells/well). After being treated by Tet for 0, 12, 24, 36, and 48 h, 50 µL MTT solution (KeyGEN BioTech, Nanjing, China) was added to each well. Four hours later, the supernatant was removed, and 150 µL DMSO (KeyGEN BioTech) was added to dissolve the formazan crystals. A microplate reader (BioTek Instruments, Winooski, VT, USA) was used to export the absorbance at the wavelength of 490 nm.

### EdU assay
Cells were seeded in 24-well plates at a density of $2 \times 10^4$ cells each well. Cells were treated with 10 µM EdU (KeyGEN BioTech) for 2 h before fixing in 4% polyformaldehyde for 15 min at room temperature. Then, cells were incubated with 0.1 mL 0.5% Triton X-100 (Beyotime) for 20 min at room temperature. After being treated with Click-iT reaction solution for 30 min in the dark, nuclei were dyed with DAPI for 5 min. Cells were captured with a fluorescence microscope (400 × , Olympus, Tokyo, Japan).

### Cloning formation assay
Cells were digested and resuspended, and 300 cells were seeded into a petri dish. The medium was changed every 3 d. Two weeks later, the cell colonies were stained with Giemsa complex dye (Nanjing Jiancheng Bioengineering Institute, Nanjing, China) for 5 min. The number of clones was counted and the cloning formation rate was calculated: cloning formation rate (%) = (amount of clones/number of inoculated cells) × 100%.

### Flow cytometry detection
Apoptosis was detected by flow cytometry with Annexin V- FITC/PI staining. Briefly, cells were seeded into 6-well plates at a density of $5 \times 10^5$ cells each well. After infection or transfection, cells were harvested, washed twice with PBS, and resuspended in 500 µL binding buffer. Then, cells were stained with 5 µL Annexin V-FITC (KeyGEN BioTech) and 5 µL PI (KeyGEN BioTech) for 10 min in the dark, and analyzed by flow cytometry (Aceabio, San Diego, CA, USA).

### Subcutaneous xenografts of nude mice

All animal experimental procedures were approved by the Ethical Committee of Shengjing Hospital of China Medical University (No. 2022PS983K). Male BALB/c nude mice (16 ± 1 g, 4 weeks) were purchased from ChangZhou Cavens Experimental Animal Co., Ltd (Jiangsu, China). After one week of adaptive feeding, mice were randomly divided into 4 groups: pLVX-EV + Tet, pLVX-KCTD15 + Tet, pLKO-shNC + Tet, and pLKO-shKCTD15 + Tet. HCT116 or LoVo cells ($5 \times 10^6$ cells) were injected subcutaneously into the nude mice. When tumors were visible, KCTD15 overexpression or knockdown was induced by feeding mice with 1 mg/mL Tet in drinking water. The diameters of each tumor mass were measured every 4 d to obtain the tumor growth curve. After 28 days, all mice were sacrificed and xenografted tumors were collected. We have complied with all relevant ethical regulations for animal use.

### IHC assay

The paraffin-embedded tumor tissues were cut into 5-μm sections, then deparaffinized and rehydrated. Sections were incubated in the antigen repair solution for 10 min and treated with 3% $H_2O_2$ for 15 min at room temperature to eliminate endogenous peroxidase activity. After being blocked with 1% bovine serum albumin (BSA, Sangon Biotech, Shanghai, China) for 15 min at room temperature, sections were treated with primary antibody (p53: 1:100 dilution, Affinity, China; Ki67: 1:100 dilution, Affinity) at 4 °C overnight followed by the incubation of horseradish peroxidase (HRP)-labeled secondary antibody (1:500 dilution) for 60 min at 37 °C. After being stained with 3,3'-Diaminobenzidine (DAB, Fuzhou Maixin Biotechnology Development Co., Ltd., Fuzhou, China) and hematoxylin (Solarbio), sections were dehydrated, treated with neutral gum, and mounted with coverslips (400 ×, Olympus).

### TUNEL staining

After being deparaffinized and rehydrated, 5-μm tumor sections were treated with 50 μL 0.1% Triton X-100 for 8 min at room temperature. Then, the sections were incubated with TUNEL reaction solution (50 μL, enzyme solution: label solution = 1: 9) for 60 min in the dark at 37 °C. After being stained with DAPI for 5 min in the dark, images were observed under a fluorescence microscope (400 ×, Olympus).

### RIP assay

An EZ-Magna RIP RNA-Binding Protein Immunoprecipitation Kit (Millipore) was used. Briefly, cells were harvested and lysed by the RIP lysis buffer. To prepare the magnetic beads, 5 μg FTO antibody or IgG was preincubated with magnetic beads to capture magnetic beads-antibody complex. Then, the supernatant (100 μL) of cell lysate was immunoprecipitated with the complex overnight at 4 °C. After that, the beads were washed six times with 500 μL RIP washing buffer and resuspended in 150 μL protease K. The co-immunoprecipitated RNA was purified using phenol: chloroform: isoamyl alcohol for PCR or qPCR analysis.

### MeRIP assay

MeRIP was performed by using a riboMeRIPTM m6A Transcriptome Profiling Kit (Ribobio, Guangzhou, China). The RNAs were isolated and fragmented in RNA fragmentation buffer (contained in the Kit). The anti-m6A antibody (5 μg) or IgG (5 μg) was conjugated to protein A/G magnetic beads. The fragmented RNAs were immunoprecipitated with the complex of antibody-protein A/G beads for 2 h at 4 °C. The modified RNA was eluted by competition with free m6A and recovered using the Magen Hipure Serum/plasma miRNA Kit. Next, the enrichment of mRNA with m6A modification was examined by qRT-PCR.

### Dual-luciferase reporter assay

KCTD15 sequences containing the high confidence m6A sites (sequence 1: from 1100 to 1324 bp, sequence 2: from 2080 to 2265 bp) were inserted into a pmir-GLO dual luciferase expression vector (General Biosystems, Anhui, China) containing Renilla luciferase (R-luc) and firefly luciferase (F-luc) to construct a wild-type KCTD15 reporter plasmid. For mutation, the third base A within the m6A motif was replaced by a base C. The wild-type or mutant KCTD15 reporter vectors were co-transfected with FTO overexpression vectors into HCT116 cells. After 48 h, luciferase activity was detected by a microplate reader (Biotek). The F-luc/R-luc activity ratio was calculated.

### Co-IP and ubiquitination assay

Methods of protein extraction and concentration detection were described in the Western Blot section. Co-IP was performed using a PierceTM immunoprecipitation kit (Thermo Scientific) according to the manufacturer's instructions. p53 or HDAC1 primary antibody (10 μg each) was first cross-linked with AminoLink coupling resin. The isolated proteins were then incubated with antibody-crosslinked resin for 2 h at room temperature. The precipitated protein was eluted with elution buffer and detected by Western Blot. The antibody details were provided in Supplementary Table. S4.

### Statistics and reproducibility

Data were analyzed using GraphPad Prism 8.0.1 (GraphPad Software Inc., San Diego, CA, USA) software. Student's t-test or one-way analysis of variance (ANOVA) was used to analyze the differences between groups. The results were presented as means and the error bars represented the standard deviation. $P < 0.05$ was considered statistically significant. All experiments were performed at least three times. The number of samples per independent experiment were described in the legends.

### Reporting summary

Further information on research design is available in the Nature Portfolio Reporting Summary linked to this article.

## Data availability

The source data underlying the graphs in this study are available at Supplementary Data 1. The uncropped blot images are provided in Supplementary Figs. 4–10. All other data are available from the corresponding author on reasonable request.

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

## Author contributions

Fang-Yuan Zhang and Lin Wu designed the study, performed experiments, analyzed the results, and wrote the manuscript; Tie-Ning Zhang and Huan-Huan Chen supervised and guaranteed the study.

## Competing interests

The authors declare no competing interests.
