## [Peer Review File · Communications Biology]

Reviewers' comments:

Reviewer #1 (Remarks to the Author):

In the manuscript of Zhang et al., the authors focus on the role of KCTD15 in the progression of colorectal cancer. They refer to the role of demethylase FTO mediated by m6A-YTHDF2 as regulators of KCTD15 mRNA levels in cancer cells. They also suggest that KCTD15 ability to inhibit CRC progression is due to its ability to increase the stability of p53 by acting on its acetylation levels.

The work is extremely interesting and clarifies the potential role of KCTD15 in carcinogenesis. The results shown support the hypotheses made by the authors. There are a few points that should be expanded and clarified to ensure the publication of the manuscript.

-I do not understand why the authors do not also use another CRC model system for the KCTD15 methylation and p53 stability analysis experiments. It would be appropriate, to corroborate the hypotheses made, to show some experiments done on HCT116, also on LoVo cells.

-It would be useful to understand whether cells over-expressing KCTD15 or silenced for KCTD15 alter their tumor aggressiveness, through specific experiments such as cell migration experiments. This is because reduced cellular aggressiveness could also support the hypothesis of KCTD15 as a therapeutic target in CRC.

-The authors also show that alterations in KCTD15 expression levels lead to the stabilization of p53 through the action of HDAC1. The authors should demonstrate whether KCTD15 is able to interact directly with p53 or with HDAC1.

Minor issues:

- Line 274. There is no reference to Figure 2D in the text.

- Line 308 It says "decreased", I guess the authors meant "Increased"

-Line 341. It says KCTD16, I guess they meant to write KCTD15.

Revise the English, some sentences are difficult to understand.

Reviewer #2 (Remarks to the Author):

In this paper, the authors claim that KCTD15 is regulated by demethylase FTA in an m6A-YTHDF2-dependent manner and exerts a tumor inhibiting role in CRC progression by increasing p53 stability. The data are interesting and a role for KCTD15 in CRC cells is clearly delineated by the authors. In particular, author's claim that KCTD15 overexpression attenuated cell proliferation in vitro and xenograft tumor growth in vivo is convincingly demonstrated.

Also, I find convincing the data regarding how m6A modification affects KCTD15 expression. Regarding the mechanism of action by KCTD15 which has been proposed, modulation of p53 by means of HDAC1 may be not the only mechanism responsible for tumor suppression, being HDAC1 also involved in modulation of the Hh pathway (by modulating Gli1 acetylation) and other targets, and it seems not fully demonstrated that p53 stabilization is the only mechanism involved in CRC antitumoral effect. I would not rule out other effects of KCTD15 on other pathways as well, which may contribute to the effect shown.

Some of the figures need improvements, especially with the addition of the appropriate controls.

On this regard, the authors should address the following:

1) The authors present in Figure 1 the DEGs. Among them, several hint (panel d and e), to a role of genes involved in muscle, muscle contraction, cardiomyocytes, cardiomyopathy. The authors should spend at least a few words discussing these findings.

2) I believe that, before focusing on KCTD15 alone, the authors should spend a few words elaborating what is known about kctd7.

3) Figure 2 Panel d: the authors should show HIC of tumor and the paired non tumoral tissue, as well, to better evaluate the level of KCTD15 expression.

4) Figure 4 Panel e, we should see also Ki67 protein levels in the WBs.

5) Figure 5:

Panel b, KCTD15 protein should be shown as well.

Panel c: is important to add a graph with the actual numbers of tunel positive cells.

Panel D the quality of the samples and of the staining is low (maybe adding in supplementary a couple more tumors would be helpful)

Panel e is not necessary, can be removed, does not add anything new.

6) Figure 6, panel F: misspelling of the word sequence

7) Figure 7 panel d: the protein levels of FTO and YTHDF1 should be shown.

8) Figure 8:

panel A: K15 protein is missing.

Panel b, p53 input is not shown

Panel c: to allow a better comparison, densitometry performed on WB with similar signal (exposures) at timepoint T0 within the EVctrl and KCTD15 samples would be useful. Otherwise, the difference in stability is not clear (i.e. the percentage of reduction of the protein with time may be similar, overall).

Panel d, instead of siTP53, it would be better to use a siHDAC1, if the mechanism is thorough HDAC1.

Discussion:

1) The authors claim that KCTD15 suppresses CRC progression via increasing protein stability of p53. The final claim is not fully supported by the data presented. It is published that KCTD15 may act through suppression of the HH pathway in other contexts, so it should be made clear that action through p53 is not the only potential mechanism.

The authors themselves admit that p53 mutations commonly occur in approximately 40-50% of CRC (line 441). So, the fact that KCTD15 is downregulated in most of the tumors, regardless of p53 status suggests involvement in other tumor suppressive mechanisms. It is possible that the p53 mechanism and the Hh pathway are somewhat acting separately or in cooperation.

1) The authors claim that KCTD15 expression was significantly downregulated in CRC tissues and was negatively associated with the TNM stage of CRC patients.

While it is clear that KCTD15 expression reduction is significant in CRC datasets, less clear is the relation with TNM stage. Indeed, in Table 1: the authors claim that KCTD15 may participate in the progression of CRC (line 62). What it is readable from the table 1 is that the correlation between K15 and TNM stage suggests that stages I-II have preferentially low K15 levels, while looking at stage II-III, (although the number of samples is low), we do not observe a significant increase in % of samples with K15 reduction compared with samples with high K15 expression. So it is possible that K15 loss plays a more significant role in early stage tumors, while this loss is not so useful in later stages. The authors should elaborate on this. Furthermore, it would be really useful to have in table1 also a correlation between K15 expression and follow-up data such as disease free or 5 years survival.

Minor:

Mistake in line 427: Spiombi et al work on medulloblastoma cells.

Reviewer #3 (Remarks to the Author):

the author found that KCTD15 expression was significantly downregulated in CRC tissues and was negatively associated with the TNM stage of CRC patients. KCTD15 overexpression attenuated cell proliferation in vitro and xenograft tumor growth in vivo, this is interesting, but some problems need further improvement.

1. There is grammatical error in row 51, it should be 'Including ... and cancers'.

2. Since KCTD7 also showed significant difference and seemed to be a good target gene, please demonstrate your reason for choosing KCTD15 but not KCTD7 as the target gene.

3. In this article, the researchers found that KCTD15 played an inhibiting role in CRC. However, in table 1, the data showed patients with lower expression of KCTD15 represented an early stage of cancer, please explain it.
4. Figures of Edu assay seems be weird. EdU and dapi/Hoechst could be incorporated into DNA while EdU incorporated into newly synthesized DNA, dapi/Hoechst incorporated into whole DNA. In this way, the extent of dyed DNA should be same. However, in figure 3b, figure 5c and figure 8e, the extents of nuclear dyed by EdU were smaller than those dyed by dapi/Hoechst.
5. In panel 8 of Fig3c, the representative figure of pLKO-shKCTD15 was not clear enough, please substitute it with a high-resolution one.
6. The histograms in Figure 5a seems the same and generated by same data. These two histograms doesn't match with the representative figures of FACS.
7. Please add the results of apoptosis assays in KCTD15-knockdown cells.
8. Please interpret the method and protocol of RNA stability assay.
9. This article showed a potential relationship between FTO and KCTD15 and FTO could mediate the protein expression of KCTD15. However, the expression change of FTO in CRC and normal tissues and the relationship between FTO and KCTD15 in clinical samples were not clarified.
10. Figure 8b should modified to be easier to understand.
11. Rescue assays should be supplemented.
12. The relationship among FTO, KCTD15 and P53 and their impact in prognosis showed be clarified.

Reviewers' comments

Reviewer #1

Remarks to the Author:

In the manuscript of Zhang et al., the authors focus on the role of KCTD15 in the progression of colorectal cancer. They refer to the role of demethylase FTO mediated by m6A-YTHDF2 as regulators of KCTD15 mRNA levels in cancer cells. They also suggest that KCTD15 ability to inhibit CRC progression is due to its ability to increase the stability of p53 by acting on its acetylation levels.

The work is extremely interesting and clarifies the potential role of KCTD15 in carcinogenesis. The results shown support the hypotheses made by the authors. There are a few points that should be expanded and clarified to ensure the publication of the manuscript.

Response: Thank you very much for the overall comments!

Major issues

1. I do not understand why the authors do not also use another CRC model system for the KCTD15 methylation and p53 stability analysis experiments. It would be appropriate, to corroborate the hypotheses made, to show some experiments done on HCT116, also on LoVo cells.

Response: Thanks to your kind suggestions, now we realize that it would be better to add results derived from LoVo cells. Several key experiments that demonstrated the regulation of KCTD15 methylation and p53 stability were additionally carried out in LoVo cells. The results were shown in Figure 6 a-b/d, Figure 7a and Figure 8a-b.

2. It would be useful to understand whether cells over-expressing KCTD15 or silenced for KCTD15 alter their tumor aggressiveness, through specific experiments such as cell migration experiments. This is because reduced cellular aggressiveness could also support the hypothesis of KCTD15 as a therapeutic target in CRC.

**Response:** We totally agree that it would be useful to investigate the role of KCTD15
in CRC by determining if it alters tumor aggressiveness. However, we believe the
reviewer also noted that the main aim of our study was to comprehensively investigate
the role of KCTD15 in CRC growth. Therefore, we further investigate how KCTD15
affected p53, a critical regulator in cancer cell survival.

CRC metastasis is a complex topic, we are also afraid that we could not reveal the
mechanisms underlying how KCTD15 affects CRC cell metastasis by merely adding
the migration assay. Please kindly understand that our group prefer to focusing on one
mechanism (CRC cell growth), instead of two (CRC cell growth and metastasis),
keeping the integrity of the present study. In fact, we are investigating KCTD15's role
in CRC aggressiveness, however, we have not obtained a solid conclusion yet. We will
absolutely share these findings in the future.

Therefore, we added this point as a limitation, and again thank you the kind
suggestion (lines 381-385).

3. The authors also show that alterations in KCTD15 expression levels lead to the
stabilization of p53 through the action of HDAC1. The authors should demonstrate
whether KCTD15 is able to interact directly with p53 or with HDAC1.

**Response:** Thanks for your comments.

Co-IP assay was performed to investigate the binding between KCTD15 and HDAC1
both in HCT116 and LoVo cells (Figure 8b). The results showed that KCTD15
downregulated HDAC1 protein expression but did not interact with HDAC1.

A previous study from Spiombi et al. ¹ demonstrated that KCTD15 reduced HDAC1
protein expression without interacting with HDAC1. Given the fact that HDAC1 is a
pivotal regulator of p53 deacetylation ², in the present study, we analyzed the protein
levels of HDAC1, total p53 and acetylated-p53 in CRC cells. We found that KCTD15
induced p53 acetylation by decreasing HDAC1 expression (Figure 8).

We have to admit that we failed to describe the precise mechanisms explaining the
regulation of KCTD15 on HDAC1. Spiombi et al. ¹ demonstated that KCTD15 induced

HDAC1 degradation by increasing KCASH2 (KCTD Containing-Cullin Adaptor 2)
expression. KCASH2 interacts with Cullin3 to form a E3 ubiquitin ligase complex,
which can recruit and degrade HDAC1³. We added this information in lines 424-428.

The major aims of our study are to: 1st) explore how KCTD15 affects CRC cell
growth and apoptosis; 2nd) whether the abnormal expression of KCTD15 in CRC tissues
is associated with m6A. Honestly, at first, we only determined the effects of KCTD15
on CRC cell apoptosis by performing Annexin V/PI staining. Since p53 is a critical
tumor suppressor, after the group discussion, we decided to analyze p53 expression post
the genetic manipulation of KCTD15, and then we were inspired by Spiombi et al.¹ to
determine the expression of HDAC1.

The reason why we did not investigate the interaction between KCTD15 and
KCASH2 in CRC cells is that we plan to perform IP-LC/MS based on high-throughput
proteomics to comprehensively explore if there are other molecules involved in
KCTD15's regulation on HDAC1. As we answered in Question 2, we preferred to not
compressing all research contents into one single study. Please kindly let us summarize
these unresolved questions into our next study.

Thank you so much again!

**Minor issues**

-
1. Line 274. There is no reference to Figure 2D in the text.

**Response:** Here, we detected the expression of KCTD15 in the CRC and para-
cancerous tissues by IHC staining, and we have added the description of Figure 2D in
the text (lines 260-262).

2. Line 308 It says "decreased", I guess the authors meant "Increased".

**Response:** So sorry for such careless clerical error. We have corrected it (lines 289-
290).

3. Line 341. It says KCTD16, I guess they meant to write KCTD15.

**Response:** So sorry for such careless clerical error. We have corrected it (line 315).

4. Revise the English, some sentences are difficult to understand.

**Response:** We have checked and revised the English of our manuscript thoroughly.

**References**

1. Spiombi, E. et al. KCTD15 inhibits the Hedgehog pathway in Medulloblastoma
cells by increasing protein levels of the oncosuppressor KCASH2. *Oncogenesis*
**8**, 64, (2019).

2. Ito, A. et al. MDM2-HDAC1-mediated deacetylation of p53 is required for its
degradation. *The EMBO Journal* **21**, 6236-6245, (2002).

3. Canettieri, G. et al. Histone deacetylase and Cullin3-REN(KCTD11) ubiquitin
ligase interplay regulates Hedgehog signalling through Gli acetylation. *Nature*
*Cell Biology* **12**, 132-142, (2010).

**Reviewer #2**

**Remarks to the Author:**

In this paper, the authors claim that KCTD15 is regulated by demethylase FTO in an
m6A-YTHDF2-dependent manner and exerts a tumor inhibiting role in CRC
progression by increasing p53 stability. The data are interesting and a role for KCTD15
in CRC cells is clearly delineated by the authors. In particular, author's claim that
KCTD15 overexpression attenuated cell proliferation in vitro and xenograft tumor
growth in vivo is convincingly demonstrated. Also, I find convincing the data regarding
how m6A modification affects KCTD15 expression.

**Response: Thank you very much for the overall comments!**

**Major issues**

-----

-

1. Regarding the mechanism of action by KCTD15 which has been proposed,
modulation of p53 by means of HDAC1 may be not the only mechanism responsible
for tumor suppression, being HDAC1 also involved in modulation of the Hh pathway
(by modulating Gli1 acetylation) and other targets, and it seems not fully demonstrated
that p53 stabilization is the only mechanism involved in CRC antitumoral effect. I
would not rule out other effects of KCTD15 on other pathways as well, which may
contribute to the effect shown.

**Response: Thank you so much for your comments, please let us explain.**

The major aims of our study are to: 1st) explore how KCTD15 affects CRC cell
growth and apoptosis; 2nd) whether the abnormal expression of KCTD15 in CRC is
associated with m6A.

Honestly, at first, we only determined the effects of KCTD15 on CRC cell apoptosis
by performing Annexin V/PI staining. Since p53 is a critical tumor suppressor, after the
group discussion, we decided to analyze p53 expression post the genetic manipulation
of KCTD15, and then we were inspired by Spiombi et al. ¹ to investigate HDAC1.
Moreover, we found that neither HDAC1 overexpression nor TP53 inhibition
completely abolished the apoptosis of CRC cells induced by KCTD15, suggesting an

involvement of other mechanisms underlying KCTD15's pro-apoptotic function. We
plan to perform high-throughput proteomics to comprehensively explore if there are
other molecules involved in KCTD15's actions in CRC cells. However, we preferred
to not compressing all research contents into one single study. Please kindly let us
summarize these unresolved questions into our next study. We have added this point as
a limitation of our study (lines 436-441).

Again, thank you so much!

2. Some of the figures need improvements, especially with the addition of the
appropriate controls.

On this regard, the authors should address the following:

2.1 The authors present in **Figure 1** the DEGs. Among them, several hint (panel d and
e), to a role of genes involved in muscle, muscle contraction, cardiomyocytes,
cardiomyopathy. The authors should spend at least a few words discussing these
findings.

**Response:** As suggested, we have described the findings from GO and KEGG
enrichment analyses in the Discussion section (lines 244-248 and 362-365).

2.2 I believe that, before focusing on KCTD15 alone, the authors should spend a few
words elaborating what is known about KCTD7.

**Response:** Thanks for the reviewer's comments. We have elaborated on the known
features of KCTD7 in the Discussion section (lines 375-380).

2.3 Figure 2 Panel d: the authors should show IHC of tumor and the paired non
tumoral tissue, as well, to better evaluate the level of KCTD15 expression.

**Response:** As suggested, we additionally detected the expression of KCTD15 in CRC
tumor and the paired non-tumoral tissues by IHC staining, and the results confirmed
the downregulated expression of KCTD15 in cancerous tissues (Figure 2d).

2.4 Figure 4 Panel e, we should see also Ki67 protein levels in the WBs.

**Response:** We additionally analyzed the Ki67 protein levels in CRC cells following
KCTD15 overexpression or knockdown (Figure 4d). The results were consistent with
IHC staining shown in Figure 4e.

2.5 Figure 5:

Panel b: KCTD15 protein should be shown as well.

**Response:** Added in Figure 5b as suggested.

Panel c: is important to add a graph with the actual numbers of tunel positive cells.

**Response:** As requested by the reviewer, we have quantified the percentage of TUNEL-
positive cells (Figure 5c).

Panel d: the quality of the samples and of the staining is low (maybe adding in
supplementary a couple more tumors would be helpful)

**Response:** Based on your suggestions, we have re-provided the results of IHC staining
in the revised version (Figure 5d).

Panel e: is not necessary, can be removed, does not add anything new.

**Response:** The Figure 5e has been deleted.

2.6 Figure 6, panel f: misspelling of the word sequence

**Response:** Corrected as suggested!

2.7 Figure 7 panel d: the protein levels of FTO and YTHDF1 should be shown.

**Response:** Added as suggested (Figure 7e).

2.8 Figure 8:

Panel a: K15 protein is missing.

**Response:** Added as suggested (Figure 8a).

Panel b, p53 input is not shown.

**Response:** We have rearranged the images (Figure 9a).

Panel c: to allow a better comparison, densitometry performed on WB with similar
signal (exposures) at timepoint T0 within the EVctrl and KCTD15 samples would be
useful. Otherwise, the difference in stability is not clear (i.e. the percentage of reduction
of the protein with time may be similar, overall).

**Response:** As suggested by the reviewer, we have quantified the protein amounts of
p53 using a normalization method with the signal at time point T0 arbitrarily set to
100%. The protein stability of p53 was evaluated by degradation curve in cells exposed
to CHX (Figure 9b). We found that the KCTD15 delayed p53 degradation by
determining the protein half-life of p53.

Panel d, instead of siTP53, it would be better to use a siHDAC1, if the mechanism is
thorough HDAC1.

**Response:** As suggested, we added results derived from the genetic manipulation of
HDAC1 in the revised version. Figure 8c-f showed that HDAC1 upregulation reversed
the alteration caused by KCTD15 in HCT116 cells.

3. Discussion

3.1 The authors claim that KCTD15 suppresses CRC progression via increasing
protein stability of p53.

The final claim is not fully supported by the data presented. It is published that KCTD15
may act through suppression of the HH pathway in other contexts, so it should be made
clear that action through p53 is not the only potential mechanism.

The authors themselves admit that p53 mutations commonly occur in approximately
40-50% of CRC (line 441). So, the fact that KCTD15 is downregulated in most of the

tumors, regardless of p53 status suggests involvement in other tumor suppressive
mechanisms. It is possible that the p53 mechanism and the Hh pathway are somewhat
acting separately or in cooperation.

**Response:** Thank you so much for reading our manuscript carefully!

TP53 mutations occur in approximately 40-50% of CRC ², and the mutant TP53 may
encode inactive p53 ³. HCT116 and LoVo cells with no TP53 mutation were used in
this study to ensure that the endogenous p53 function as an apoptotic inducer. A
previous study from Spiombi et al. ¹ demonstrated that KCTD15 reduced HDAC1
protein expression without interacting with HDAC1. Given the fact that HDAC1 is a
pivotal regulator of p53 deacetylation ⁴, we here analyzed the protein levels of HDAC1,
total p53 and acetylated-p53 in CRC cells. We found that KCTD15 induced p53
acetylation and decreased HDAC1 expression (Figure 8).

The major aims of our study are to: 1st) explore how KCTD15 affects CRC cell
growth and apoptosis; 2nd) whether the abnormal expression of KCTD15 in CRC is
associated with m6A.

Honestly, at first, we only determined the effects of KCTD15 on CRC cell apoptosis
by performing Annexin V/PI staining. Since p53 is a critical tumor suppressor, after the
group discussion, we decided to analyze p53 expression post the genetic manipulation
of KCTD15, and then we were inspired by Spiombi et al. ¹ to investigate HDAC1.
Herein, we also found that KCTD15 overexpression enhanced p53 acetylation and
upregulated its expression in a HDAC1 dependent manner. Moreover, neither HDAC1
overexpression nor TP53 inhibition completely abolished the apoptosis of CRC cells
induced by KCTD15, suggesting an involvement of other mechanisms underlying
KCTD15's pro-apoptotic function. We plan to perform high-throughput proteomics to
comprehensively explore if there are other molecules involved in KCTD15's actions in
CRC cells.

As we answered in Question 1, we preferred to not compressing all research contents
into one single study. Please kindly let us summarize these unresolved questions into
our next study.

Moreover, since we also agree that p53 pathway may not be the only downstream
effector to KCTD15, we believe that our conclusion was exaggerated. We modified the
title and the descriptions in the revised article (lines 1-2, 422-434, and 436-441).

3.2 The authors claim that KCTD15 expression was significantly downregulated in
CRC tissues and was negatively associated with the TNM stage of CRC patients.

While it is clear that KCTD15 expression reduction is significant in CRC datasets, less
clear is the relation with TNM stage. Indeed, in Table 1: the authors claim that KCTD15
may participate in the progression of CRC (line 62). What it is readable from the table
1 is that the correlation between K15 and TNM stage suggests that stages I-II have
preferentially low K15 levels, while looking at stage III, (although the number of
samples is low), we do not observe a significant increase in % of samples with K15
reduction compared with samples with high K15 expression. So it is possible that K15
loss plays a more significant role in early stage tumors, while this loss is not so useful
in later stages. The authors should elaborate on this. Furthermore, it would be really
useful to have in table 1 also a correlation between K15 expression and follow-up data
such as disease free or 5 years survival.

**Response:** We agree with the reviewer that patients from stages I-II have preferentially
low K15 levels.

We believe that you have also noted that only 10 patients from stages III were recited
in this study. With the popularization of routine health examination, it is hard to collect
enough samples from patients of late TNM stage. We will continue our study and tried
our best to collect more clinical samples to further reveal whether KCTD15 is related
to the late CRC stage. We have clarified this point in the Discussion section (lines 386-
393).

The survival data of patients involved in table 1 are still being collected. Please
kindly understand that we may not be able to add this information, and we will share
these data in the future.

To address your concern, we extracted data from online database UALCAN

(<http://ualcan.path.uab.edu>). The survival curves below showed that patients with
higher KCTD15 expression had a better prognosis. Although these data suggested a
correlation of between KCTD15 and CRC prognosis, the statistical analysis was not
insignificant. KCTD15 and its family members are the current research emphases of
our group, the clinical data are being collected, we will share these data in the future.

Moreover, in order not to lead any misunderstanding to the readers, we discussed
the present findings properly the revised manuscript (lines 393-398).

Mistake in line 427: Spiombi et al work on medulloblastoma cells.

**Response:** Modified as suggested (lines 424-426). Your careful examination impressed
300 us a lot, thank you!

References

1 Spiombi, E. et al. KCTD15 inhibits the Hedgehog pathway in Medulloblastoma
cells by increasing protein levels of the oncosuppressor KCASH2. *Oncogenesis*
**8**, 64, doi:10.1038/s41389-019-0175-6 (2019).

2 Takayama, T., Miyanishi, K., Hayashi, T., Sato, Y. & Niitsu, Y. Colorectal cancer:
genetics of development and metastasis. *Journal of gastroenterology* **41**, 185-
192, doi:10.1007/s00535-006-1801-6 (2006).

3 Wang, Z., Strasser, A. & Kelly, G. L. Should mutant TP53 be targeted for cancer
therapy? *Cell Death Differ* **29**, 911-920, doi:10.1038/s41418-022-00962-9
(2022).

4 Ito, A. et al. MDM2-HDAC1-mediated deacetylation of p53 is required for its
degradation. *EMBO J* **21**, 6236-6245 (2002).

**Reviewer #3**

**Remarks to the Author:**

The author found that KCTD15 expression was significantly downregulated in CRC
tissues and was negatively associated with the TNM stage of CRC patients. KCTD15
overexpression attenuated cell proliferation in vitro and xenograft tumor growth in vivo,
this is interesting, but some problems need further improvement.

**Response: Thank you very much for the overall comments!**

Major issues

1. There is grammatical error in row 51, it should be ‘Including ... and cancers’.

**Response: Corrected as suggested (line 50). Moreover, we checked and corrected the**
**whole manuscript carefully.**

2. Since KCTD7 also showed significant difference and seemed to be a good target
gene, please demonstrate your reason for choosing KCTD15 but not KCTD7 as the
target gene.

**Response: Thanks for the reviewer’s comments.**

**Our group are interesting in both KCTD7 and KCTD15. The present study focused**
**on KCTD15, and we will carry our further study regarding to KCTD7. We also**
**discussed the role of KCTD7 in the modified manuscript (lines 373-380).**

**According to your suggestion, to highlight KCTD15 in this study, we moved results**
**related to KCTD7 into the supplementary materials (Supplementary Figure 1).**

3. In this article, the researchers found that KCTD15 played an inhibiting role in CRC.
However, in table 1, the data showed patients with lower expression of KCTD15
represented an early stage of cancer, please explain it.

**Response: Thank you very much for pointing this out!**

**We believe that you have also noted that only 10 patients from stages III were recited**
**in this study. With the popularization of routine health examination, it is hard to collect**

enough samples from patients of late TNM stage. We will continue our study and tried
our best to collect more clinical samples to further reveal whether KCTD15 is related
to the late CRC stage.

Moreover, in order not to lead any misunderstanding to the readers, we discussed
the present findings properly the revised manuscript (lines 386-398). Your careful
examination impressed us a lot, thank you!

4. Figures of Edu assay seems be weird. EdU and dapi/Hoechst could be incorporated
into DNA while EdU incorporated into newly synthesized DNA, dapi/Hoechst
incorporated into whole DNA. In this way, the extent of dyed DNA should be same.
However, in figure 3b, figure 5c and figure 8e, the extents of nuclear dyed by EdU were
smaller than those dyed by dapi/Hoechst.

**Response:** Thanks for pointing this out. Based on your kind suggestion, we re-
performed the EdU assay, but the results were consistent with the results presented
earlier. Then we did some survey in this area and found the below information.

EdU is a thymidine nucleoside analogue that can replace nucleobase T during cell
proliferation. EdU labeling can accurately label proliferative cells. DAPI is an
embedding agent for DNA containing a specific AT sequence, and it adheres to the
minor groove region of DNA double helix. Therefore, is there a possibility that these
two cannot be totally overlapped? We found some results similar to ours (see red ovals
below).

Again, thank you very much for your carefulness. In our future study, we will utilize
Edu kit from multiple manufacturers, and share these findings.

1. PMID: 32994691

2. PMID: 33784010

5. In panel 8 of Fig3c, the representative figure of pLKO-shKCTD15 was not clear
 enough, please substitute it with a high-resolution one.

**Response:** Replace as suggested in panel 8 of Figure 3c.

6. The histograms in Figure 5a seems the same and generated by same data. These two
 histograms doesn't match with the representative figures of FACS.

**Response:** The histograms showed the mean values from three replicates. To
 demonstrate that the histograms in Fig. 5a are not generated from the same data, we
 provided the original FCS files from three replicates in the supplementary materials.

Please see “Original FCS files (for review)”.

7. Please add the results of apoptosis assays in KCTD15-knockdown cells.

**Response:** Thanks for your comments. As suggested, we performed the apoptosis
assays after knocking down KCTD15, and found no significant alteration in cell
apoptosis. The number of apoptotic cells in normal CRC cell population is very small.
Related description was added (lines 285-287).

8. Please interpret the method and protocol of RNA stability assay.

**Response:** Please see lines 143-144.

9. This article showed a potential relationship between FTO and KCTD15 and FTO
could mediate the protein expression of KCTD15. However, the expression change of
FTO in CRC and normal tissues and the relationship between FTO and KCTD15 in
clinical samples were not clarified.

**Response:** We highly appreciate your constructive suggestion!

As suggested, we additionally detected the expression of FTO in clinical samples
from CRC patients and used Pearson analysis to analyze whether it was correlated with
KCTD15. We found that their expression was positively correlated (Supplementary
Figure 3; $r = 0.84$; $P < 0.0001$).

10. Figure 8b should be modified to be easier to understand.

**Response:** Modified as suggested (revised Figure 9a).

11. Rescue assays should be supplemented.

**Response:** Added as suggested!

In the updated version, we have assessed the effects of simultaneous overexpression
of KCTD15 and HDAC1 on cellular behaviors. The results showed that the forced
expression of HDAC1 partly reversed alterations caused by KCTD15 overexpression

(Figure 8c-f).

12. The relationship among FTO, KCTD15 and P53 and their impact in prognosis
showed be clarified.

**Response:** FTO was downregulated in CRC tissues and plays an anti-tumor role in
CRC cells through its m6A demethylase activity ^{1,2}. p53 is a classical suppressor in
varied cancers, including CRC ³.

Our study revealed the following issues:

a. Lower mRNA and protein expression of KCTD15 in the tumor tissues from CRC
patients.

b. FTO induced m6A de-methylation, leading to the upregulation of KCTD15.

c. KCTD15 inhibited CRC cell growth and induced apoptosis, partly by activating anti-
tumor p53 pathway.

We sincerely apologize for not describe their relationship clearly in the original text.

We now added more details in the revised version (lines 70-89).

**References**

1 Ruan, D.-Y. et al. FTO downregulation mediated by hypoxia facilitates
colorectal cancer metastasis. *Oncogene* **40**, 5168-5181, doi:10.1038/s41388-
021-01916-0 (2021).

2 Relier, S. et al. FTO-mediated cytoplasmic m6Am demethylation adjusts stem-
like properties in colorectal cancer cell. *Nature Communications* **12**, 1716,
doi:10.1038/s41467-021-21758-4 (2021).

3 Schulz-Heddergott, R. et al. Therapeutic Ablation of Gain-of-Function Mutant
p53 in Colorectal Cancer Inhibits Stat3-Mediated Tumor Growth and Invasion.
*Cancer Cell* **34**, doi:10.1016/j.ccell.2018.07.004 (2018).

REVIEWERS' COMMENTS:

Reviewer #1 (Remarks to the Author):

I thank the authors for answering all my suggestions precisely. They carried out new experiments and added the limitations that had emerged.

The work, already very explanatory, has been greatly improved with several new experiments and precise explanations of previous experiments. Congratulations on adding an important piece for the scientific community in clarifying the functional role of the KCTD15 protein.

Reviewer #2 (Remarks to the Author):

I believe the authors addressed most of my concerns, significantly improving their paper, that now is suitable for publication.

Reviewer #3 (Remarks to the Author):

No more comments